# FOSP: Fine-tuning Offline Safe Policy through World Models

**Chenyang Cao**[1]**, Yucheng Xin**[1]**, Silang Wu**[1]**, Longxiang He**[1]**, Zichen Yan**[2]**, Junbo Tan**[1]*****,
**Xueqian Wang**[1]*****
[1] Shenzhen International Graduate School, Tsinghua University,
[2] College of Design and Engineering, National University of Singapore

## Abstract

Offline Safe Reinforcement Learning (RL) seeks to address safety constraints by learning from static datasets and restricting exploration. However, these approaches heavily rely on the dataset and struggle to generalize to unseen scenarios safely. In this paper, we aim to improve safety during the deployment of vision-based robotic tasks through online fine-tuning an offline pretrained policy. To facilitate effective fine-tuning, we introduce model-based RL, which is known for its data efficiency. Specifically, our method employs in-sample optimization to improve offline training efficiency while incorporating reachability guidance to ensure safety. After obtaining an offline safe policy, a safe policy expansion approach is leveraged for online fine-tuning. The performance of our method is validated on simulation benchmarks with five vision-only tasks and through real-world robot deployment using limited data. It demonstrates that our approach significantly improves the generalization of offline policies to unseen safety-constrained scenarios. To the best of our knowledge, this is the first work to explore offline-to-online RL for safe generalization tasks. The videos are available at `https://sunlighted.github.io/fosp_web/`.

## 1 Introduction

Offline Reinforcement Learning (RL) has been widely studied within the academic community (Wu et al., 2019; Fujimoto et al., 2019; Kostrikov et al., 2021; Kumar et al., 2020), using pre-collected datasets to learn policies without exploration (Levine et al., 2020). At the offline training stage, safety constraints can be incorporated into policy learning, which focuses on learning to be safe rather than learning safely. In this way, offline trained policy is promising to avoid constraint violations during online deployment. However, offline datasets cannot fully simulate complicated real-world environments and will cause out-of-distribution (OOD) issues in the case of unseen data (Kumar et al., 2020). Such generalization tasks are significant challenges for offline learning and safe learning.

This paper aims to explore the design of a practical safe RL algorithm that can both leverage offline data and quickly adapt to novel environments while maintaining safety properties. A straightforward way is to train a safe RL policy on offline data. However, the offline trained policy is prone to distribution shift issues, making it hard to apply to unseen environments due to severe safety constraint violations. Hence, a safe offline-to-online training mechanism is necessary to improve generalization. We consider utilizing model-based methods to complete generalization as quickly as possible. Model-based RL has proven to be a powerful tool to enhance sample efficiency by constructing a world model of the environment (Hafner et al., 2019a; Cang et al., 2021; Hafner et al., 2019b; 2020; 2023). The world model reduces random explorations by predicting the outcomes of different actions, leading to a fast convergence speed during online fine-tuning after minimal interactions (Feng et al., 2023). Moreover, it significantly improves action safety in high-dimensional tasks such as vision-based operations due to its prediction ability (As et al., 2022; Huang et al., 2023). Thus, we leverage its advantages to address the safe generalization problem in real-world deployment.

---

*Corresponding authors

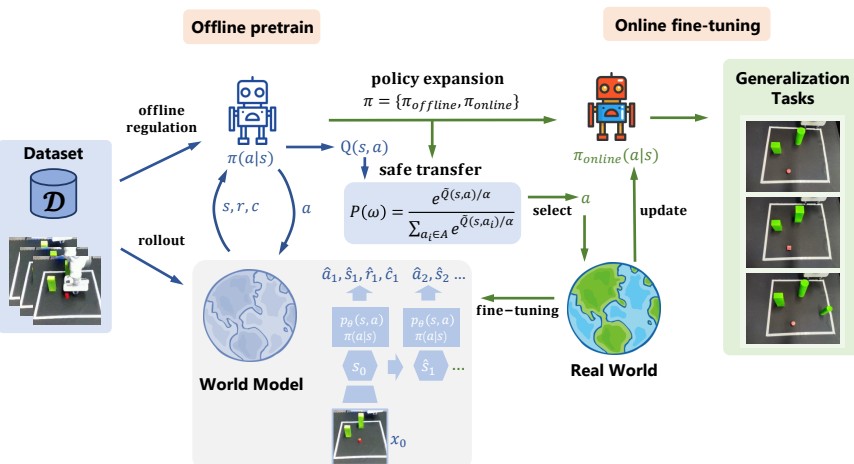

*Figure 1.* **Fine-tuning offline safe policy through world models.** We propose a framework for offline pretraining and online fine-tuning the world model. We first pretrain the agent by the offline dataset and rollouts generated from world models. The grey section depicts the architecture of the world model: it first encodes an image observation into its latent state $s_0$, then, for each latent state, generates an action using the policy, as well as predicts the reward, cost, and next state. In the offline-to-online phase, we employ policy expansion to initialize a new policy for online fine-tuning. The pretrained Q-value is leveraged to construct a softmax probability distribution. Then, we select an action by this distribution for the agent to safely interact with the real world, generalizing it to novel tasks.

We consider a framework of pretraining an offline safe policy with a world model and then directly fine-tuning it during online interactions. However, such a strategy usually causes degradation of policy performance and a lack of safety after fine-tuning (Nair et al., 2020). The main reason lies in too many mixed constraints (both soft and hard), such as safety constraints and behavior regularization, making it difficult to ensure both optimality and safety. Meanwhile, the world model will introduce prediction errors due to the distribution shift during the fine-tuning on unseen tasks, leading to increasing safety risks. Therefore, we seek to handle mixed constraints and use the offline safe policy as guidance for online correction. Based on this key insight, we propose **FOSP** (**F**ine-tuning **O**ffline **S**afe **P**olicy through World Models), which enhances offline safe training and bridges model-based offline training with online fine-tuning without constraints violations, as shown in Figure 1.

Taking advantage of SafeDreamer (Huang et al., 2023), a safe version of DreamerV3 (Hafner et al., 2023), we employ world models to optimize both offline and online RL, significantly improving data utilization efficiency. Specifically, during offline pretraining, our approach leverages in-sample optimization to update the Q-value conservatively in order to deal with the value overestimation. We also consider safe constraints with a feasibility-guided method, while simultaneously using the reachability estimation function to address the mixed constraints involved in this problem. Moreover, we adopt the safe policy expansion (Zhang et al., 2023) to bridge offline and online algorithms, avoiding performance drops during the initial stage and suboptimality with nearly zero violations. In this way, FOSP balances the trade-off between optimal performance and constraint violations, allowing for safe fine-tuning on generalization tasks.

To evaluate our method, we design experiments across numerous Safety-Gymnasium (Ji et al., 2023) tasks and real-world robotic arms including offline training and online fine-tuning. For simulation tasks, experimental results show robust performance in both offline and online fine-tuning phases and achieve nearly zero cost, outperforming prior RL algorithms. Furthermore, we deploy FOSP in real-world experiments, utilizing a Franka manipulator to perform trajectory planning tasks. It turns out that our method's ability across tasks with different safety regions can be transferred to unseen safety-critical scenarios through few-shot fine-tuning.

We summarize our contributions as follows: (1) To the best of our knowledge, FOSP is the **first approach** that tackles safe generalization tasks by offline-to-online RL. (2) It handles the trade-off between the performance and constraints satisfaction across offline and online training phases on

various vision-only tasks. (3) In real-world deployment, it does **not need sim-to-real transfer** and can be safely fine-tuned in **unseen safety-constrained scenarios**. (4) It can solve offline-to-online safe RL adaptation by **only a few trials** while maintaining safety.

## 2 RELATED WORKS

**Offline-to-online RL** Offline RL, which trains a policy by leveraging a large amount of existing data, is valuable to real-world scenarios (Wu et al., 2019; Fujimoto et al., 2019). Prior offline RL works focus on addressing the distribution shift and value overestimation problem (Yu et al., 2020; 2021; Rafailov et al., 2021; Kumar et al., 2020; Kidambi et al., 2020; Fujimoto & Gu, 2021). Recently, in-sample methods have demonstrated their robust performance by avoiding out-of-distribution (OOD) state-action pairs (Kostrikov et al., 2021; Xu et al., 2023; Garg et al., 2023; Peng et al., 2019; Nair et al., 2020; He et al., 2024; Yang et al., 2024). Unlike training solely on offline data, offline-to-online RL involves further fine-tuning the offline policy online. And many offline RL algorithms have extended to online fine-tuning (Kostrikov et al., 2021; Nair et al., 2020). But these methods have shown that the more effective the offline RL method is, the worse online fine-tuning performance is (Xiao et al., 2023). Some previous works have explored how to effectively fine-tune from offline pretrained policy, including balanced sampling in replay buffer (Lee et al., 2022b), policy calibrating (Nakamoto et al., 2024; Rudner et al., 2021), and parameter transferring (Xie et al., 2021; Rajeswaran et al., 2017). However, existing solutions are restricted by policy transitions or the need to estimate the density of the online policy distribution (Zhao et al., 2022). PEX (Zhang et al., 2023) is an approach that bridges offline and online algorithms and is adaptive for fine-tuning different online methods with offline policy guidance. We extend it to safe algorithms that not only improve the performance but also ensure safety, for it is shown as an effective approach in offline-to-online training.

**Safe RL** Safe RL focuses on optimizing objectives with different constraints (Altman, 2021). Many methods are based on policy search algorithms that ensure adherence to near-constraints iteratively (Achiam et al., 2017) and Lagrangian methods that convert it to an unconstrained problem to solve (Chow et al., 2018; Ding et al., 2020; Tessler et al., 2018). Model-based RL methods utilize world models for action selection so that they ensure safety by background planning. Some recent works are trying to address challenges remaining in vision-input tasks (Sikchi et al., 2022; Hansen et al., 2022; Hafner et al., 2019b). LAMDBA (As et al., 2022) is a method extending DreamerV1 (Hafner et al., 2019a) to safe RL by using augmented Lagrangian. Safe-SLAC (Hogewind et al., 2022) incorporating the Lagrangian method into SLAC (Lee et al., 2020) improves the computation complexity of LAMBDA. SafeDreamer (Huang et al., 2023) is the latest work in safe model-based reinforcement learning. It enhances safety in online or background planning with DreamerV3 (Hafner et al., 2023), achieving state-of-the-art (SOTA) performance. However, prior works only aim to address issues in the online setting, which cannot avoid costs incurred during exploration in the real world. By contrast, some safe offline RL can learn policies from offline datasets by using OOD action detection (Cao et al., 2024; Xu et al., 2022; Li et al., 2022), DICE-based theory (Lee et al., 2022a), and sequence modeling (Lin et al., 2023; Liu et al., 2023). Nonetheless, they only utilize soft constraints which sometimes can not effectively decrease constraint violations. Some works consider the hard constraints by using control theory methods such as control barrier function (CBF) (Choi et al., 2020) and HJ reachability (Zheng et al., 2023; Yu et al., 2023; Ganai et al., 2024; Fisac et al., 2019). Our work draws from these works and unifies all the constraints with model-based background planning, continuing the CMDP paradigm that satisfies safety constraints in expectation.

## 3 PRELIMINARIES

**World Model** Our work extends DreamerV3 (Hafner et al., 2023), a powerful and robust baseline model-based RL using world model and actor-critic. It has the ability to model high-dimensional observations of the environment and increase the data efficiency of RL. The world model learns compressed image input representations and predicts future states and rewards sequences for decision-making. It is implemented as a Recurrent State-Space Model (RSSM) (Hafner et al., 2019b). First, the model maps the high-dimensional observations $x_t$ to stochastic states $z_t$. And deterministic latent states $h_t$ are predicted by $z_t$ and actions $a_t$. By concatenating both of them, we get the model state $s_t = \{h_t, z_t\}$. Finally, we facilitate $s_t$ to predict the next observations and rewards. Due

to the safety considerations in the environment, we add a cost decoder to the original model. All components in the world model are as follows:

latent encoder:   $z_t \sim q_\theta(z_t \mid h_t, x_t)$    observation decoder:   $\hat{x}_t \sim p_\theta(x_t \mid z_t, h_t)$

deterministic state:   $h_t = f_\theta(z_{t-1}, h_{t-1}, a_{t-1})$    reward decoder:   $\hat{r}_t \sim p_\theta(r_t \mid z_t, h_t)$

stochastic state:   $\hat{z}_t \sim p_\theta(z_t \mid h_t)$    cost decoder:   $\hat{c}_t \sim p_\theta(c_t \mid z_t, h_t)$

The model is trained end-to-end by the ELBO or variational free energy of a hidden Markov model loss function with KL balancing:

$$\mathcal{L}^{\text{model}}(\theta) = \mathbb{E}_{\tau \sim \mathcal{D}}\Big[ \sum_{t=1}^{T} -\ln p_\theta(x_t \mid s_t) - \ln p_\theta(r_t \mid s_t) - \ln p_\theta(c_t \mid s_t) +$$

$$\mathbb{D}_{KL}[sg(q_\theta(z_t|h_t, x_t))||p_\theta(z_t|h_t)] + \beta \mathbb{D}_{KL}[q_\theta(z_t|h_t, x_t)||sg(p_\theta(z_t|h_t))]\Big]. \tag{1}$$

The stop-gradient operator $sg(\cdot)$ makes the representations more predictable in the imagination training.

**Safe Model-based RL**   Safe RL is frequently formulated as a Constrained Markov Decision Process (CMDP) $\mathcal{M} = (\mathcal{S}, \mathcal{A}, \mathbb{P}, \mathcal{R}, \mathcal{C}, \mu, \gamma)$ (Altman, 2021) with discrete time steps $t \in \{0, \ldots, T\}$. $\mathcal{S}$ and $\mathcal{A}$ denote the state and action space. $\mathbb{P}(s'|s, a)$ represents the transition probability from $s'$ to $s$ under action $a$. $\mathcal{R}$ is reward space that maps $\mathcal{S} \times \mathcal{A}$ to $\mathbb{R}$. And $\mathcal{C}$ is a cost function set containing cost functions $c : \mathcal{S} \times \mathcal{A} \to [0, C_{max}]$. $\mu(\cdot) : \mathcal{S} \to [0, 1]$ is the initial state distribution and $\gamma \in (0, 1)$ is the discount factor. Given a policy distribution $\pi_\psi$, the cost state value function $V^c$, and cost action-state value function $Q_c$ are defined by $V^c(s) = \mathbb{E}_{\tau \sim \pi_\psi}[\sum_{t=0}^{T} \gamma^t c(s_t, a_t)], Q^c(s, a) = \mathbb{E}_{\tau \sim \pi_\psi}[\sum_{t=0}^{T} \gamma^t c(s_t, a_t)]$ like the common MDP. We define the finite-horizon reward function and cost function as follows:

$$J^{\mathcal{R}}(\pi_\psi) = \mathbb{E}_{a_t \sim \pi_\psi, s_{t+1} \sim p_\theta, s_0 \sim \mu}\left[ \sum_{t=0}^{T} r_t \Big| s_0 \right], \tag{2}$$

$$J_i^{\mathcal{C}}(\pi_\psi) = \mathbb{E}_{a_t \sim \pi_\psi, s_{t+1} \sim p_\theta, s_0 \sim \mu}\left[ \sum_{t=0}^{T} c_t^i \Big| s_0 \right] \leqslant d^i, \ \forall i \in \{1, \ldots, C\}, \tag{3}$$

where $i$ denotes different safety constraints we want to avoid and $d^i$ are cost thresholds. Therefore, our safe model-based RL problem can be formulated as follows:

$$\max_{\psi} J^{\mathcal{R}}(\pi_\psi) \ \text{ s.t. } \ J_i^{\mathcal{C}}(\pi_\psi) \leqslant d^i, \ \forall i \in \{1, \ldots, C\}. \tag{4}$$

**Reachability Estimation Function**   RESPO (Ganai et al., 2024) first introduced the reachability estimation function to capture the probability of constraint violations. Suppose that the set $\mathcal{S}_v$ is a constraint violation set defined in the state space $\mathcal{S}$, containing all states that violate the constraints. The reachability estimation function (REF) $u^\pi : \mathcal{S} \to [0, 1]$ can be defined as:

$$u^\pi(s) := \mathbb{E}_{\tau \sim \pi}\Big[ \max_{s_t \in \tau} \mathbb{1}\{(s_t|s_0, \pi) \in \mathcal{S}_v\} \Big]. \tag{5}$$

The value $\max_{s_t \in \tau} \mathbb{1}\{(s_t|s_0, \pi) \in \mathcal{S}_v\}$ is defined on specific trajectory. It equals 1 if existing violations and 0 otherwise.

## 4 METHODS

In this section, we introduce FOSP for fine-tuning safe actions with offline pretrained world models. According to Figure 1, we first train an offline safe policy and subsequently fine-tune it online. In Section 4.1, we address the Q-value overestimation in offline learning by utilizing in-sample optimization to train the critic network $Q_\phi(s, a)$. In Section 4.2, to balance the trade-off between performance and constraint violations during offline training, we employ the reachability estimation function to train the safe actor $\pi_\psi(s)$. In Section 4.3, the policy is online fine-tuned by safe policy expansion, which helps mitigate the performance drop and enables further improvements. The whole algorithm can be found in Appendix C.

## 4.1 IN-SAMPLE OPTIMIZATION FOR OFFLINE TRAINING

Value overestimation is an extensive challenge in offline RL due to the distribution shift between the state-action pairs from the dataset and those generated by the learned policy. This issue becomes even more pronounced in model-based offline RL algorithms, where both the policy and a dynamics model $p_\theta$ are learned from the offline data. It not only amplifies value overestimation but also causes additional complications in estimating the latent dynamics and predicting future returns, further impacting the overall performance. To mitigate these challenges, we propose the in-sample optimization as a solution.

First, we consider a sequence of data $\mathcal{B} = (\boldsymbol{x}_{1:T}, \boldsymbol{a}_{1:T}, \boldsymbol{r}_{1:T}, \boldsymbol{c}_{1:T})$ sampled from the offline dataset. Then, the model generates latent states $\boldsymbol{s}_{1:T}^0 \sim q_\theta(\boldsymbol{s}_{1:T}|\boldsymbol{h}_{1:T}, \boldsymbol{x}_{1:T})$, which are used as initial states $\hat{\boldsymbol{s}}_{1:T}^0$ to generate rollouts:

$$\hat{\boldsymbol{a}}_{1:T}^h \sim \pi_\psi(\boldsymbol{a}|\hat{\boldsymbol{s}}_{1:T}^h), \ \hat{\boldsymbol{s}}_{1:T}^{h+1} \sim p_\theta(\boldsymbol{s}|\hat{\boldsymbol{a}}_{1:T}^h, \hat{\boldsymbol{s}}_{1:T}^h), \ \hat{\boldsymbol{r}}_{1:T}^h \sim p_\theta(\boldsymbol{r}|\hat{\boldsymbol{s}}_{1:T}^h), \ \hat{\boldsymbol{c}}_{1:T}^h \sim p_\theta(\boldsymbol{c}|\hat{\boldsymbol{s}}_{1:T}^h), \quad (6)$$

where $h$ represents the horizon of the generated sequence. We use $Q_\phi(\boldsymbol{s}, \boldsymbol{a})$ as the critic network. Motivated by Implicit Q-learning (IQL) (Kostrikov et al., 2021), we only use in-sample actions as calculated components of $\lambda$-return. Due to out-of-distribution actions from $\pi_\psi(\boldsymbol{s})$, a high Q-value from $Q_\phi$ doesn't always mean the action will lead the agent to a desirable state. Thus, a separate value network $V_\varphi(\boldsymbol{s}) = \mathbb{E}_{\boldsymbol{a} \sim \pi_\psi(\cdot|\boldsymbol{s})}[Q_\phi(\boldsymbol{s}, \boldsymbol{a})]$ only conditioned on states is proposed to mitigate the out-of-distribution issue. It approaches the action distribution expectation of Q-value and can be optimized by expectile regression:

$$\mathcal{L}_V(\varphi) = \mathbb{E}_{\tau \sim \mathcal{D}, \pi_\psi, p_\theta}\Big[|\kappa - \mathbb{1}\{Q_\phi(\hat{\boldsymbol{s}}_t^j, \hat{\boldsymbol{a}}_t^j) - V_\varphi(\hat{\boldsymbol{s}}_t^j) < 0\}|(Q_\phi(\hat{\boldsymbol{s}}_t^j, \hat{\boldsymbol{a}}_t^j) - V_\varphi(\hat{\boldsymbol{s}}_t^j))^2\Big], \quad (7)$$

where $\kappa \in (0, 1)$ is a constant. We use rollouts in equation 6 to estimate returns by TD($\lambda$) and update the critic by equation 9.

$$R_t^H = V_\varphi(\hat{\boldsymbol{s}}_t^H), \ R_t^h = \hat{\boldsymbol{r}}_t^h + \gamma((1 - \lambda)V_\varphi(\hat{\boldsymbol{s}}_t^h) + \lambda R_t^{h+1}), \quad (8)$$

$$\mathcal{L}_Q^1(\phi) = -\frac{1}{HT}\mathbb{E}_{\tau \sim \mathcal{D}, \pi_\psi, p_\theta}\Big[\sum_{t=1}^T \sum_{h=1}^H (Q_\phi(\hat{\boldsymbol{s}}_t^h, \hat{\boldsymbol{a}}_t^h) - R_t^h)^2\Big], \quad (9)$$

where $H$ is the horizon of the model and $h \leqslant H$. It is noted that the $\lambda$-return in equation 9 is estimated solely through the generated rewards. However, during the early stages of offline training, the world model may not be able to generate accurate rewards. The real rewards from the offline dataset can better guide the critic's learning process. Therefore, we add equation 10 as a regularization term to the critic loss to improve learning efficiency:

$$\mathcal{L}_Q^2(\phi) = -\frac{1}{T}\mathbb{E}_{\tau \sim \mathcal{D}, \pi_\psi, p_\theta}\Big[\sum_{t=1}^T [Q_\phi(\boldsymbol{s}_t^0, \boldsymbol{a}_t) - (\boldsymbol{r}_t + \gamma V_\varphi(\boldsymbol{s}_{t+1}^0))]^2\Big], \quad (10)$$

where $\boldsymbol{a}_t, \boldsymbol{r}_t$ are from the offline dataset and the final critic loss becomes

$$\mathcal{L}_Q(\phi) = \mathcal{L}_Q^1(\phi) + \mathcal{L}_Q^2(\phi), \quad (11)$$

## 4.2 REACHABILITY ESTIMATION FUNCTION AS SAFETY GUARANTEE

To ensure the safety of the policy, prior works consider adopting the relaxation of the Lagrangian multiplier method (Stooke et al., 2020). The optimization problem on equation 4 can be transformed into an unconstrained problem with the Lagrangian multiplier $\lambda_i$:

$$\max_{\pi_\psi} \min_{\lambda_i \geqslant 0} J^\mathcal{R}(\pi_\psi) - \lambda_i(J_i^\mathcal{C}(\pi_\psi) - d^i), \ \forall i \in \{1, \dots, C\}. \quad (12)$$

However, the soft constraints will still lead to a certain chance of violation, which is exacerbated in the offline setting due to the difficulty in estimating the cost values. Hence, we replace soft constraints with hard constraints by the feasibility guidance approach (Yu et al., 2022). The feasible states are included in the feasible set $\mathcal{S}_f(\boldsymbol{s}) := \{\boldsymbol{s}|V_\varphi^c(\boldsymbol{s}) = 0\}$. Then we can divide the optimization problem into two parts: the feasible part and the infeasible part.

For the feasible part, we want the agent to maintain safety within the feasible set and maximize the reward return. For the infeasible part, we want the policy to violate the constraints as little as

possible and return to the feasible set. As the safe offline RL problem has another objective to optimize: $D_{\mathrm{KL}}(\pi_\psi||\pi_b)$ ($\pi_b$ is the behavior policy), we can reformulate the problem as follows:

$$\textbf{Feasible part:} \quad \max_{\pi_\psi} \mathbb{E}_s\Big[V_\varphi^r(s)\cdot\mathbb{1}\{s\in\mathcal{S}_f\}\Big], \quad \text{s.t.} \quad V_\varphi^c(s)=0, \quad D_{\mathrm{KL}}(\pi_\psi||\pi_b)\leqslant\epsilon, \quad (13)$$

$$\textbf{Infeasible part:} \quad \max_{\pi_\psi} \mathbb{E}_s\Big[-V_\varphi^c(s)\cdot\mathbb{1}\{s\notin\mathcal{S}_f\}\Big], \quad \text{s.t.} \quad D_{\mathrm{KL}}(\pi_\psi||\pi_b)\leqslant\epsilon. \quad (14)$$

Note that the policy constraint $D_{\mathrm{KL}}(\pi_\psi||\pi_b)\leqslant\epsilon$ is a soft constraint, which complicates the feasible part problem due to coupled hard and soft constraints. Thus, we use the reachability estimation function $u^\pi(s)=\mathbb{E}_{\tau\sim\pi}[\max_{s_t\in\tau}\mathbb{1}\{(s_t|s_0,\pi)\in\mathcal{S}_v\}]$ from RESPO (Ganai et al., 2024), where $\mathcal{S}_v$ and $\mathcal{S}_f$ are complementary. This function helps handle the mixed constraints by prioritizing the hard constraints and also assists the agent in reentering the feasible region even when it ventures into the infeasible region. And the problem becomes:

$$\max_{\pi_\psi} \mathbb{E}_s\Big[V_\varphi^r(s)\cdot(1-u_\psi^\pi)-V_\varphi^c(s)\cdot u_\psi^\pi\Big], \quad \text{s.t.} \quad V_\varphi^c(s)=0, \quad D_{\mathrm{KL}}(\pi_\psi||\pi_b)\leqslant\epsilon. \quad (15)$$

To simplify the constraints, we introduce Augmented Lagrangian with a proximal relaxation method. Meanwhile, we leverage advantage functions $A^r(s,a)=Q_\phi^r(s,a)-V_\varphi^r(s)$ and $A^c(s,a)=Q_\phi^c(s,a)-V_\varphi^c(s)$ to assess the influence of actions on the value function according to Proposition 1. The final problem can be formulated as follows:

$$\max_{\pi_\psi} \mathbb{E}_s\Big[(A^r(s,a)-\Phi(V_\varphi^c(s),\lambda_p^k,\mu^k))\cdot(1-u_\psi^\pi)-A^c(s,a)\cdot u_\psi^\pi\Big], \quad \text{s.t.} \quad D_{\mathrm{KL}}(\pi_\psi||\pi_b)\leqslant\epsilon. \quad (16)$$

$$\Phi\left(V_\varphi^c,\lambda_p^k,\mu^k\right)=\begin{cases}\lambda_p^k V_\varphi^c+\frac{\mu^k}{4}(V_\varphi^c)^2 & \text{if } \lambda_p^k+\frac{\mu^k}{2}V_\varphi^c\geqslant 0,\\ -\frac{(\lambda_p^k)^2}{\mu^k} & \text{otherwise .}\end{cases} \quad (17)$$

where $\lambda_p^k$ and $\mu^k$ are Lagrange multipliers. We can finally obtain a closed-form solution of $\pi_\psi$, which we can use to extract the optimal policy (Zheng et al., 2023). (For more details see Appendix.A.2)

$$\pi^*(a|s)=\frac{1}{Z}w\cdot\pi_b(a|s), \quad (18)$$

$$\text{where} \quad w=(1-u_\psi^\pi)\cdot\exp(\beta_1 A^r(s,a))+u_\psi^\pi\cdot\exp(-\beta_2 A^c(s,a)). \quad (19)$$

and the policy loss is as follows:

$$\mathcal{L}_\pi(\psi)=-\mathbb{E}_{\tau\sim\mathcal{D}}\Big[\sum_{t=1}^T(w\cdot\log\pi_\psi(a_t|s_t)+\eta E[\pi_\psi(a_t|s_t)])-\Phi(V_\varphi^c(s),\lambda_p^k,\mu^k)\cdot(1-u_\psi^\pi)\Big], \quad (20)$$

where $\eta$, $\beta_1$, $\beta_2$ are hyperparameters, $E$ is the entropy. For critic learning, we use the same way to train cost critic $Q_{\phi'}^c(s,a)$ and cost value $V_{\varphi'}^c(s)$ as Section 4.1.

## 4.3 SAFE POLICY EXPANSION FOR WORLD MODELS FINE-TUNING

We aim to improve performance and generalize the learned policy to similar tasks while maintaining the safety learned from the offline policy through online fine-tuning. However, applying online safe RL algorithms for fine-tuning often leads to initial performance drops and constraint violations, probably caused by the distribution shift between the offline and online data (Lee et al., 2022b). Conversely, directly fine-tuning the original offline RL algorithm typically exhibits worse online performance due to their conservative design (Kostrikov et al., 2021; Kumar et al., 2020). Therefore, we try to bridge offline-online RL by safe policy expansion.

Suppose that $\pi_\psi$ is the offline learned policy. Rather than directly fine-tuning it in the online stage, we optimize a new policy $\pi_{\psi'}$ by gradually learning from frozen prior policy $\pi_\psi$. Concretely, given a current state $s$, an action set from which actions are selected is composed of actions generated by two policies: $\mathcal{A}=\{a_\psi\sim\pi_\psi(s),a_{\psi'}\sim\pi_{\psi'}(s)\}$. The actions are then probabilistically selected based on their potential utilization value (Zhang et al., 2023):

$$P(\omega)[k]=\frac{\exp(\tilde{Q}(s,a_k)/\alpha)}{\exp(\tilde{Q}(s,a_\psi)/\alpha)+\exp(\tilde{Q}(s,a_{\psi'})/\alpha)}, \quad \forall k\in\{\psi,\psi'\}, \quad (21)$$

*Table 1.* **Offline-to-online results.** Reward return and cost return of methods in offline-to-online fine-tuning (1M steps for offline and 0.5M steps for online fine-tuning). We report the mean value of 5 independent runs with different seeds.

| | FOSP(Ours) | | Recover-RL | | DreamerV3 | | SafeDreamer | |
|---|---|---|---|---|---|---|---|---|
| Tasks | Reward ↑ | Cost ↓ | Reward ↑ | Cost ↓ | Reward ↑ | Cost ↓ | Reward ↑ | Cost ↓ |
| PointGoal1 | **18.7**→21.5 | **1.4**→0.2 | 5.4→12.6 | 1.8→0.8 | 17.4→**25.8** | 82.3→85.1 | 12.7→19.2 | 1.5→**0.02** |
| PointButton1 | 14.7→18.1 | **9.5**→**2.1** | 5.4→6.3 | 10.3→2.8 | **15.1**→**20.8** | 167.7→157.9 | 8.9→10.7 | 10.2→4.5 |
| PointPush1 | **4.0**→13.2 | 18.1→**0.1** | 0.4→3.8 | 24.8→0.5 | 2.3→**14.6** | 30.9→26.3 | 1.3→10.3 | **17.8**→**0.12** |
| PointGoal2 | 8.1→13.5 | **7.5**→**0.23** | 0.5→2.7 | 53.2→0.3 | **9.5**→**18.9** | 367.1→290.2 | 3.6→12.7 | 8.6→0.3 |
| CarGoal2 | 10.1→14.5 | 1.6→**0.07** | 4.6→8.9 | **0.6**→**0.13** | 16.7→**24.1** | 231.0→287.9 | 6.9→9.8 | 0.7→0.09 |
| Average | 16.6 | **0.54** | 6.7 | 0.91 | **20.8** | 171.1 | 12.5 | 1.01 |

*Table 2.* **Compare with online algorithms.** We compare our method with some online-only algorithms. SafeDreamer(+planning) was trained online for 0.5M steps and CPO and PPO-Lagrangian were trained for 10M training steps until they converged. We report the mean value of 5 independent runs with different seeds.

| | FOSP(Ours) | | SafeDreamer(+planning) | | CPO | | PPO-Lag | |
|---|---|---|---|---|---|---|---|---|
| Tasks | Reward ↑ | Cost ↓ | Reward ↑ | Cost ↓ | Reward ↑ | Cost ↓ | Reward ↑ | Cost ↓ |
| PointGoal1 | 21.5 | **0.2** | 20.1 | 0.6 | **22.3** | 45 | 19.5 | 29 |
| PointButton1 | **18.1** | **2.1** | 12.4 | 5.5 | 17.1 | 76 | 5.3 | 25 |
| PointPush1 | **13.2** | **0.1** | 8.1 | 0.7 | 3.7 | 35 | 2.9 | 26 |
| PointGoal2 | 13.5 | **0.23** | 13.0 | 1.7 | **13.8** | 51 | 1.8 | 30 |
| CarGoal2 | 14.5 | **0.07** | 11.2 | 0.34 | **15.5** | 52 | 7.8 | 28 |
| Average | **16.6** | **0.54** | 12.9 | 1.77 | 14.5 | 51.8 | 7.5 | 27.7 |

$$\tilde{Q}(\boldsymbol{s}, \boldsymbol{a}) = \frac{Q_\phi(\boldsymbol{s}, \boldsymbol{a})}{\hat{Q}^*} - \frac{Q_{\phi'}^c(\boldsymbol{s}, \boldsymbol{a})}{\hat{C}^*}, \tag{22}$$

where $\alpha$ is temperature. We use upper bounds $\hat{Q}^* = \sum_{t=0}^{T} \gamma^t r_{max}$, $\hat{C}^* = \sum_{t=0}^{T} \gamma^t c_{max}$ to normalize the Q-value, where $r_{max}, c_{max}$ represent the max reward and cost. $\tilde{Q}(\boldsymbol{s}, \boldsymbol{a})$ represents the balance between the normalized cumulative rewards and cumulative costs. Based on the probability function, the two policies can adaptively explore the environment safely according to their respective probabilities, determined by the Q-values of their generated actions. We can ensure the stability of initial performance during the early fine-tuning stages, exploring with a better policy under the premise of minimal constraint violations. At the same time, we leverage online algorithms to enhance performance further. According to Section 4.2, the weight of policy loss in the online stage derived from equation 16 without the behavior policy constraint can be written as:

$$w = (1 - u_\psi^\pi)\beta_1 A^r(\boldsymbol{s}, \boldsymbol{a}) - u_\psi^\pi \beta_2 A^c(\boldsymbol{s}, \boldsymbol{a}). \tag{23}$$

This approach learns a new policy through online updates while leveraging the offline-trained policy as prior knowledge. Ultimately, this enables the policy to generalize to new tasks during online fine-tuning safely. In this way, the offline replay buffer can be directly converted into an online replay buffer, allowing online exploration trajectories to be continuously added during online fine-tuning, thereby simplifying the complexity of balanced sampling (Lee et al., 2022b).

# 5 EXPERIMENTAL RESULTS

In this section, we evaluate our method across different agents and tasks within the Safety-Gymnasium simulation environment (Ji et al., 2023), as well as on real-world motion planning control tasks using a Franka robot. We aim to address the following issues: (1) How does FOSP perform in offline pretrain and online fine-tuning? (2) What role does each component of FOSP play in its overall functionality? (3) How does the offline dataset affect experimental results? (4) Can FOSP successfully handle unseen safety regions in the new scenarios?

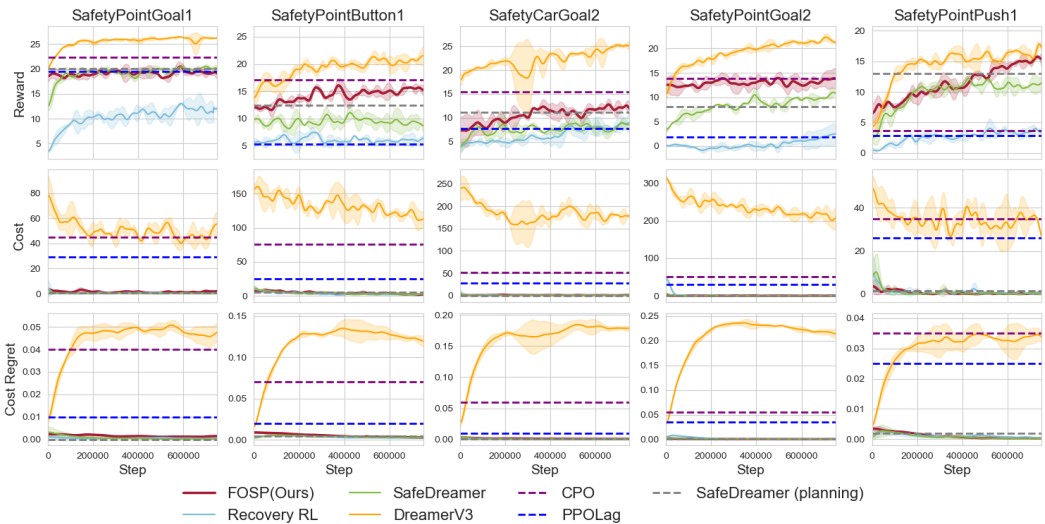

*Figure 2.* **Online experimental results.** Comparing FOSP to baselines across five image-based safety tasks at the online fine-tuning stage. The results for model-based algorithms are obtained after fine-tuning for 750,000 steps. The dashed lines represent the benchmark results for CPO and PPO-Lagrangian after 10 million training steps across all tasks. The SafeDreamer (planning) was trained online for 0.75 million steps. Reward: averaged episode reward return. Cost: averaged episode cost return. Cost Regret: averaged cost value throughout the training phase.

## 5.1 Experimental Setup

**Simulation Tasks**   We consider five tasks on Safety-Gymnasium benchmark (Ji et al., 2023) environments. The agents need to navigate to predetermined goals without collision with hazards and vases. We measure the performance with three metrics: average undiscounted episode reward return (Reward), average undiscounted episode cost return (Cost) and average cost value throughout the training phase (Cost Regret). To evaluate the advantage of world models in handling high-dimensional features, we use $64 \times 64$ pixels RGB image as inputs obtained from the first-person perspective of the agent, which is more representative of real-world environments. The standard offline datasets are sampled by three different behaviors: unsafe policy, safe policy and random policy. They are mixed in a 1:1:1 ratio, with each part containing 200 trajectories. All the tasks are trained for five seeds. For more information see Appendix.D.

**Baselines**   Prior works in offline safe RL using the model-free method typically perform poorly in vision-only tasks due to their slow convergence rate (Liu et al., 2023; Zheng et al., 2023). Therefore, we mainly compare FOSP with the following model-based baselines: **Recovery RL** (model-based version) (Thananjeyan et al., 2021), a method first learns the constraint violation regions offline and then performs online policy training; **SafeDreamer** (Huang et al., 2023), an online powerful algorithm in safe model-based reinforcement learning and **DreamerV3** (Hafner et al., 2023), a strong baseline in visual tasks but overlooks constraint violations. We also choose classical safe RL algorithms **CPO** (Achiam et al., 2017) and **PPO-Lagrangian** (Schulman et al., 2017) to compare at the online stage, following the experimental protocol of Ray et al. (2019).

**Real Robot**   The real robot environment includes a 7-DOF Franka Emika Panda robot as the controlled agent and a static third-person Intel RealSense camera to obtain images as inputs. The agent also receives additional inputs, specifically the pose information of the robot's end-effector. We design the task *SafeReach* that controls the robot to reach a desired goal while avoiding collisions with predefined obstacles in its field of view. The rewards and costs are sparse and detected manually, determining whether the robot reaches the goal or violates the constraints. The dataset consists of demonstration data, violation data, and failure data. Furthermore, we design some unseen safe regions and different targets that are not contained in the offline dataset. These unseen tasks require the agent to recognize objects that it has not encountered in the dataset and to generalize its

*Table 3.* **Real-world unseen tasks.** We record the success rate (SR, %) and the constraint violation rate (CV, %) over 20 tests in three tasks while it will be labeled as a violation if it collides with an obstacle. The robot has fine-tuned 40 gradient steps. See Appendix.E for more details.

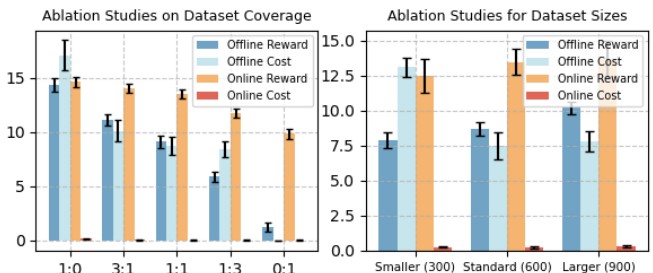

*Figure 4.* **Dataset ablation studies.** We evaluate ablations on SafetyPointGoal2 with five seeds. Due to the large offline costs affecting the visual presentation, we reduced them by 10 times in the left figures. The x label in the left image represents safe data : unsafe data. 1M steps for offline and 0.5M steps for online fine-tuning.

|       | Before FT |      | During FT |      | After FT |      |
|-------|-----------|------|-----------|------|----------|------|
| Tasks | SR ↑      | CV ↓ | SR ↑      | CV ↓ | SR ↑     | CV ↓ |
| 1     | 40        | 50   | 45        | 40   | **65**   | **20** |
| 2     | 30        | 50   | 35        | 45   | **60**   | **30** |
| 3     | 20        | 60   | 40        | 40   | **50**   | **20** |

decision-making abilities to these new scenarios. The tasks further illustrate the practicality of our approach.

## 5.2 MAIN RESULTS

We show our main offline pretrain and online fine-tuning results in Table 1, 2 to answer the question (1). During offline pretraining, FOSP outperformed the similar safe algorithms Recover RL and SafeDreamer, excelling in decreasing costs and better balancing task performance with constraint avoidance. Furthermore, it has comparable or slightly lower performance than DreamerV3 with nearly zero violations. (Appendix.H) The learning curves of online fine-tuning are shown in Figure 2. It proves that our algorithm can also be fine-tuned to achieve better performance. Meanwhile, FOSP further reduces the costs during the online fine-tuning. We also compare our method with the online planning version of Safe-Dreamer (OSRP) (Huang et al., 2023) which can only be trained online. We notice that the SafeDreamer training exclusively online outperforms the offline-to-online approach but has more constraint violations. This may be because the safety constraints learned during offline pretraining provide guidance for online fine-tuning, making the model more sensitive to safety constraints. Taking this advantage, FOSP is able to bridge offline and online safety while maintaining its performance.

## 5.3 ABLATION STUDIES

**Module Ablation** In this section, we compare the effects of different components of the model on its performance in Figure 3 to answer the question (2). Specifically, we provide the following ablations: (i) learning value function without in-sample optimization (equation 7) in Section 4.1; (ii) directly fine-tuning the model without safe policy expansion (equation 21) in Section 4.3; (iii) update safe actor without reachability estimation function in Section 4.2; (iv) design the penalty without the Augmented Lagrangian and use the feasible value function (Fisac et al., 2019). The results show that each component of our method individually (i) improves offline performance, (ii) stabilizes the transition phase from offline to online, (iii) enhances overall performance while maintaining safety constraints, and (iv) ensures compliance with safety constraints.

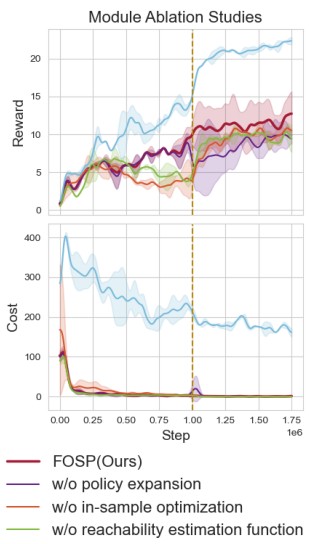

*Figure 3.* **Module ablation studies.** We evaluate ablations in SafetyPointGoal2 with means of five seeds. The vertical line divides the offline and online phases.

**Dataset Ablation** We also seek to investigate how the coverage and size of offline datasets affect the algorithm's performance to answer the question (3). For the dataset coverage, it is well established that higher rewards in the dataset lead to better performance (Yu et al., 2020). Therefore,

we focus on examining the impact of cost by adjusting the proportion of safe and unsafe data. The results in Figure 4 demonstrate that safe data tends to guide the final policy towards being overly aggressive, achieving high rewards but frequently violating constraints. On the other hand, unsafe data causes the policy to become overly conservative, hindering exploration to avoid constraint violations. This behavior arises due to errors in the dynamics model in model-based methods. When the dataset is imbalanced, the dynamics model trained through supervised learning struggles to accurately predict the costs associated with each state-action pair in the real environment. As a result, the actor-critic, which depends heavily on data generated by the dynamics model, fails to predict future state costs accurately, resulting in a suboptimal policy. To examine the impact of dataset size, we scaled the standard dataset up and down proportionally. The final results indicate that while increasing the dataset size leads to the learning of better and safer policies. Furthermore, the gains during the online fine-tuning stage are relatively modest.

### 5.4 Safe Generalization Tasks

In this section, we show how our method solves different tasks with unseen safety regions in the offline dataset and answer the question (4). We first evaluate it in the simulation, offline pretraining the agent using data collected in the *SafetyPointGoal1*, then place it in the *SafetyPointGoal2*, which has more types and a greater number of unsafe regions, for online fine-tuning. Although these tasks are the same type, the limitations of the previous data still pose significant challenges to the effectiveness and safety of fine-tuning. The experimental results in Figure 5 demonstrate the advantages of FOSP in these safety generalization tasks, showing better performance and safer fine-tuning compared to SafeDreamer.

In the real-world environment, robots will encounter many safety constraints while performing various tasks. Our key challenge is how to safely train the robots to adapt to unseen constraints by safe learning-based methods. Following this insight, we introduce obstacles and targets of different shapes and sizes in *Reach* tasks, where the robot will only encounter a limited number of these during offline pretraining. We first train the agent for 0.5 million steps using a precollected dataset, then deploy the trained agent to the real-world environment to execute tasks and fine-tune it. During online fine-tuning, each time the robot finishes a trajectory, we add it to the replay buffer and update model parameters. Our goal is to make the robot use a few trials to learn planning policies in environments with new obstacles as safely as possible. Thus, we design three real-world transfer tasks and show their results in Table 3. For more details see Appendix E.

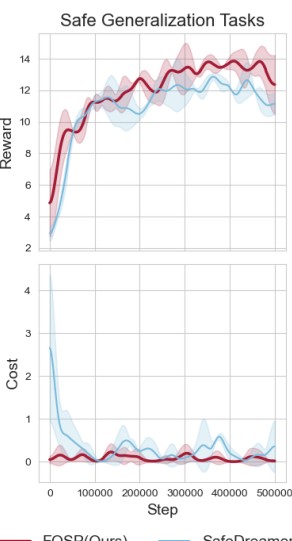

*Figure 5.* **Simulation generalization task.** We compared the performance of models pretrained 1M steps on SafetyPointGoal1 using FOSP and SafeDreamer, evaluating their fine-tuning and generalization results after 0.5M steps on SafetyPointGoal2.

In brief, FOSP can successfully finish these tasks safely, demonstrating its strong offline learning and adjustment capabilities. In contrast, only learning from the offline dataset makes it difficult to generalize to new scenarios. Its strong adaptability in fine-tuning showcases the robustness and great potential of FOSP.

## 6 Conclusion

We introduce FOSP, a novel method that integrates model-based reinforcement learning with offline-to-online fine-tuning to enhance safety in robotic tasks. FOSP addresses the challenges of balancing performance and safety during both offline and online phases using the reachability estimation function to unify different constraints and offline safe policies to guide online exploration. Our experimental results demonstrate that FOSP achieves robust performance in various safe planning tasks with vision inputs. Furthermore, we evaluate FOSP on unseen safety-critical tasks in both simulation and dynamic real-world environments, highlighting its practical value in safe few-shot fine-tuning. Overall, FOSP is the first approach to extend MBRL into a safe offline-to-online framework to solve the safe generalization problem, showing promising results in the Safety-Gymnasium benchmark.

ACKNOWLEDGMENTS

This work was supported by the National Natural Science Foundation of China (No.62103225), the Natural Science Foundation of Guangdong Province (No.2024A1515010003) and the Natural Science Foundation of Shenzhen (No.JCYJ20230807111604008, No. JCYJ20240813112007010).

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

## A  THEORETICAL INTERPRETATIONS

### A.1  DERIVATION OF THE FINAL OPTIMIZATION PROBLEM

This section analyzes the derivation from equation 13, equation 14 to equation 16 by using the Augmented Lagrangian and advantage function. First, we introduce Proposition 1, which refers to Zheng et al. (2023) (Theorem 1), Peng et al. (2019) (Appendix A).

**Proposition 1.** *The optimization objective of equation 13 is the necessary condition of* $\max_\pi \mathbb{E}_{\boldsymbol{s}}[A^r(\boldsymbol{s}, \boldsymbol{a}) \cdot \mathbb{1}\{s \in \mathcal{S}_f\}].$

*Proof.* We start with the advantage function estimation in Kakade & Langford (2002) and Peng et al. (2019). Suppose that $d_\pi(\boldsymbol{s}) = \sum_{t=0}^{T} \gamma^t p(\boldsymbol{s}_t = \boldsymbol{s}|\pi)$ is discounted state distribution with $\pi$, and $p(\boldsymbol{s}_t = \boldsymbol{s}|\pi)$ is the probability of the state $\boldsymbol{s}$ guided by $\pi$ for $t$ steps. The discounted sum of advantage expectation under $d_\pi(\boldsymbol{s})$ can be represented as follows:

$$
\mathbb{E}_{\boldsymbol{s}_{0:T} \sim d_\pi(\boldsymbol{s})} \left[ \sum_{t=0}^{T} \gamma^t A_\mu(\boldsymbol{s}_t, \boldsymbol{a}_t) \right]
$$

$$
= \mathbb{E}_{\boldsymbol{s}_{0:T} \sim d_\pi(\boldsymbol{s})} \left[ \sum_{t=0}^{T} \gamma^t \left( r(\boldsymbol{s}_t, \boldsymbol{a}_t) + \gamma V_\mu(\boldsymbol{s}_{t+1}) - V_\mu(\boldsymbol{s}_t) \right) \right]
$$

$$
= \mathbb{E}_{\boldsymbol{s}_{0:T} \sim d_\pi(\boldsymbol{s})} \left[ -V_\mu(\boldsymbol{s}_0) + \sum_{t=0}^{T} \gamma^t r(\boldsymbol{s}_t, \boldsymbol{a}_t) \right]
$$

$$
= -\mathbb{E}_{\boldsymbol{s}_0 \sim d_\pi(\boldsymbol{s}_0)} \left[ V_\mu(\boldsymbol{s}_0) \right] + \mathbb{E}_{\boldsymbol{s}_{1:T} \sim d_\pi(\boldsymbol{s})} \left[ \sum_{t=0}^{T} \gamma^t r(\boldsymbol{s}_t, \boldsymbol{a}_t) \right]
$$

$$
= -J(\mu) + J(\pi),
$$

where $\mu$ is the behavior policy used to sample data. So we get the objective representation $J(\pi) = \mathbb{E}_{\boldsymbol{s} \sim d_\pi(\boldsymbol{s})} \left[ \mathbb{E}_{\boldsymbol{a} \sim \pi(\boldsymbol{a}|\boldsymbol{s})}[A_\mu(\boldsymbol{s}_t, \boldsymbol{a}_t)] \right] + J(\mu)$. Then, we have the following derivation:

$$
\max_\pi \mathbb{E}_{\boldsymbol{s} \sim d_\pi(\boldsymbol{s})}[A^r_\pi(\boldsymbol{s}_t, \boldsymbol{a}_t)] \Rightarrow \max_\pi \mathbb{E}_{\boldsymbol{s} \sim d_\pi(\boldsymbol{s})} \left[ \mathbb{E}_{\boldsymbol{a} \sim \pi(\boldsymbol{a}|\boldsymbol{s})}[A^r_\mu(\boldsymbol{s}_t, \boldsymbol{a}_t)] \right]
$$

$$
= \max_\pi \mathbb{E}_{\boldsymbol{s} \sim d_\pi(\boldsymbol{s})} \left[ \mathbb{E}_{\boldsymbol{a} \sim \pi(\boldsymbol{a}|\boldsymbol{s})}[A^r_\mu(\boldsymbol{s}_t, \boldsymbol{a}_t)] \right] + J(\mu)
$$

$$
= \max_\pi \ J(\pi)
$$

$$
= \max_\pi \ \mathbb{E}[V^r_\pi(\boldsymbol{s}_t)].
$$

The last equation holds by the definition of $V^r_\pi(\boldsymbol{s}_t)$. □

Due to the symmetry of $\max_\pi \mathbb{E}[V^r_\pi(\boldsymbol{s})]$ and $\max_\pi \mathbb{E}[V^c_\pi(\boldsymbol{s})]$, the same proof extends to $\max_\pi \mathbb{E}[V^c_\pi(\boldsymbol{s})]$ as well. So we have proposition 2.

**Proposition 2.** *The optimization objective of equation 14 is the necessary condition of* $\max_\pi \mathbb{E}_{\boldsymbol{s}}[-A^c(\boldsymbol{s}, \boldsymbol{a}) \cdot \mathbb{1}\{s \notin \mathcal{S}_f\}].$

The problem (equation 13, equation 14) converts to:

$$
\textbf{Feasible part:} \ \ \max_{\pi_\psi} \mathbb{E}_{\boldsymbol{s}} \left[ A^r(\boldsymbol{s}, \boldsymbol{a}) \cdot \mathbb{1}\{s \in \mathcal{S}_f\} \right], \ \ \text{s.t.} \ V^c_\varphi(\boldsymbol{s}) = 0, \ \ D_{\mathrm{KL}}(\pi_\psi || \pi_b) \leqslant \epsilon, \tag{24}
$$

$$
\textbf{Infeasible part:} \ \ \max_{\pi_\psi} \mathbb{E}_{\boldsymbol{s}} \left[ -A^c(\boldsymbol{s}, \boldsymbol{a}) \cdot \mathbb{1}\{s \in \mathcal{S}_f\} \right], \ \ \text{s.t.} \ D_{\mathrm{KL}}(\pi_\psi || \pi_b) \leqslant \epsilon. \tag{25}
$$

However, it is hard for us to get a close form of $\pi_\psi$. Therefore, we begin by disregarding the constraints of the behavior policy and focus on addressing the hard safety constraints. The Augmented Lagrangian is used to deal with the feasible part (As et al., 2022). We use the following relaxation:

$$
\max_{\pi_\psi} \min_{\lambda_p \geqslant 0} \left[ A^r(\boldsymbol{s}, \boldsymbol{a}) \mathbb{1}\{s \in \mathcal{S}_f\} - \lambda_p V^c_\varphi(\boldsymbol{s}) + \frac{1}{\mu^k}(\lambda_p - \lambda_p^k)^2 \right], \tag{26}
$$

where $\lambda_p$ is a Lagrange multiplier and $\mu^k$ is a non-decreasing penalty term corresponding to gradient step $k$. By smoothing the left-hand side term in equation 24, the last term ensures that $\lambda_p$ stays near its previous estimate. Differentiating equation 24 with respect to $\lambda_p^k$ leads to the following update rule for the Lagrange multiplier:

$$\lambda_p^{k+1} = \begin{cases} \lambda_p^k + \frac{\mu^k}{2} V_\varphi^c & \text{if } \lambda_p^k + \frac{\mu^k}{2} V_\varphi^c \geqslant 0, \\ 0 & \text{otherwise .} \end{cases} \tag{27}$$

where the $\lambda_p^{k+1}$ only update when $\pi_\psi$ satisfies the constraints. The feasible objective becomes the following form:

$$J(\pi_\psi, \lambda_p^k, \mu^k) = A^r(\boldsymbol{s}, \boldsymbol{a}) \mathbb{1}\{\boldsymbol{s} \in \mathcal{S}_f\} - \Phi\left(V_\varphi^c, \lambda_p^k, \mu^k\right), \tag{28}$$

$$\Phi\left(V_\varphi^c, \lambda_p^k, \mu^k\right) = \begin{cases} \lambda_p^k V_\varphi^c + \frac{\mu^k}{4}(V_\varphi^c)^2 & \text{if } \lambda_p^k + \frac{\mu^k}{2} V_\varphi^c \geqslant 0, \\ -\frac{(\lambda_p^k)^2}{\mu^k} & \text{otherwise .} \end{cases} \tag{29}$$

Then, we utilize the reachability estimation function to connect the two parts of the problem like 15. So the final problem becomes equation 16.

## A.2 EXTRACTION OF THE OPTIMAL POLICY

We introduce a proposition from Nair et al. (2020) to solve the soft constraints from offline training to provide a solution of equation 16.

**Proposition 3.** *Suppose that the optimizing problem has the following form:*

$$\max_\pi \mathbb{E}_{\boldsymbol{a} \sim \pi}[A(\boldsymbol{s}, \boldsymbol{a})] \ \text{s.t.} \ D_{\mathrm{KL}}(\pi_\psi || \pi_b) \leqslant \epsilon.$$

*The optimal solution will satisfy:*

$$\pi^*(\boldsymbol{a}|\boldsymbol{s}) \propto \exp(\alpha A(\boldsymbol{s}, \boldsymbol{a})) \pi_b(\boldsymbol{a}|\boldsymbol{s}).$$

*Proof.* The Lagrange function can be formulated as follows:

$$L(\pi_\psi, \lambda, \mu) = \mathbb{E}_{\boldsymbol{a} \sim \pi}[A(\boldsymbol{s}, \boldsymbol{a})] - \lambda(D_{\mathrm{KL}}(\pi_\psi || \pi_b) - \epsilon),$$

Then we take the partial derivative with respect to $\pi$ and set it to 0:

$$\frac{\partial L}{\partial \pi_\psi} = A(\boldsymbol{s}, \boldsymbol{a}) - \lambda \log \pi_b(\boldsymbol{a}|\boldsymbol{s}) + \lambda \log \pi_\psi(\boldsymbol{a}|\boldsymbol{s}) = 0.$$

So we have:

$$\pi^*(\boldsymbol{a}|\boldsymbol{s}) = \frac{1}{Z} \exp(\alpha A(\boldsymbol{s}, \boldsymbol{a})) \pi_b(\boldsymbol{a}|\boldsymbol{s}),$$

where $Z$ is a normalizing constant. □

Therefore, we have a solution to equation 16. The solved $\pi_\psi$ can be explicitly expressed by equation 20.

## B IMPLEMENTATION DETAILS

To implement the REF $u_\xi$, we use the theorem introduced by Ganai et al. (2024) (Theorem 1). The REF can be trained as follows:

$$u(\boldsymbol{s}_t) = \max\{\mathbb{1}\{s \in \mathcal{S}_f\}, \gamma_u u(\boldsymbol{s}_{t+1})\}, \tag{30}$$

where $\gamma_u$ is a discount parameter $0 \ll \gamma_u < 1$ to ensure convergence of $u(\boldsymbol{s})$. Meanwhile, we ensure that its learning rate is greater than that of the critic and policy, but less than that of the Lagrangian multiplier updates.

## C    PSEUDO CODE

---

**Algorithm 1** FOSP: Fine-tuning Offline Safe Policy through World Models

---

**Require:** Offline dataset $\mathcal{D}$, policy $\pi_\psi$, critics $Q_{\phi_1}$ and cost critics $Q^c_{\phi_2}$, value network $V_{\varphi_1}, V^c_{\varphi_2}$, reachability function network $u_\xi$, world model $p_\theta$, policy rollout length $H$, number of offline training steps $N_{\text{offline}}$, number of online fine-tuning steps $N_{\text{online}}$.

1: Initialize neural network parameters $\theta, \psi, \phi_k, \varphi_k, \xi$. ($k = 1, 2$)
2: **for** $i = 1, 2, 3, \cdots, N_{\text{offline}}$ **do**
3:     Sample a batch of trajectories $\mathcal{B} \sim \mathcal{D}$.
4:     Compute model state $\boldsymbol{s}_t \sim p_\theta(\boldsymbol{s}_t|\boldsymbol{s}_{t-1})$
5:     Update $\theta$ using equation 1.
6:     Generate $H$-step latent policy rollouts using world model $p_\theta$.
7:     Compute target estimation $R_t$ by TD($\lambda$).
8:     Update $V_{\varphi_1}, V^c_{\varphi_2}$ using equation 7.
9:     Update $Q_{\phi_1}, Q^c_{\phi_2}$ using equation 11.
10:    Update $\pi_\psi$ using equation 20.
11:    Update $u_\xi$ using equation 30.
12: Initialize a new policy $\pi_{\psi'}$ and freeze offline policy $\pi_\psi$, transfer critics and values.
13: **for** $i = 1, 2, 3, \cdots, N_{\text{online}}$ **do**
14:    Switch a policy $\pi$ according to equation 21 and rollout the policy in the environment for an episode to collect a new trajectory $\tau$.
15:    $\mathcal{D} = \mathcal{D} \cup \tau$.
16:    **for** each training step **do**
17:        Sample a batch of trajectories from $\mathcal{D}$.
18:        Update $\theta$ using equation 1.
19:        Generate $H$-step latent policy rollouts using world model $p_\theta$.
20:        Compute target estimation $R_t$ by TD($\lambda$).
21:        Update $V_{\varphi_1}, V^c_{\varphi_2}$ using equation 7.
22:        Update $Q_{\phi_1}, Q^c_{\phi_2}$ using equation 11.
23:        Update $\pi_{\psi'}$ using equation 23.
24:        Update $u_\xi$ using equation 30.

---

## D    EXPERIMENTAL DETAILS

### D.1    REAL WORLD EXPERIMENTS

The experiments verify the effectiveness of the algorithm by controlling a robotic arm to avoid obstacles and complete a Reach task to the target object, called *SafeReach*. As shown in Figure 7(c), the main hardware setup consists of three parts: a Franka Panda robotic arm, an experimental platform for placing obstacles and target objects, and an Intel RealSense D435 camera for perception. On the experimental platform, we used white lines to designate a 25cm×25cm operational area as the task space for the robotic arm.

**SafeReach**    For the task of reaching, the robot receives the image information acquired by the camera along with the current joint posture information as planning conditions to determine the movement decision at the next step. As shown in Figure 7, the robotic arm needs to navigate within the designated movement space, bypassing the obstacles which are represented by the green rectangular blocks, to touch the predefined target which is represented by the red square block.

**Dataset Collection**    To train the model for the SafeReach task, 200 motion trajectories were collected as a dataset by teleoperating the robotic arm using a 3Dconnexion SpaceMouse. The dataset consists of 120 trajectories where the robotic arm did not touch any obstacles and successfully reached the target object; 30 trajectories

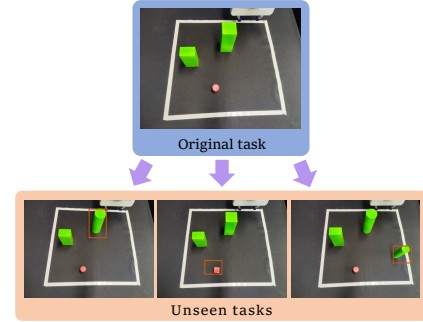

*Figure 6.* **Unseen Tasks.** We choose three unseen tasks to show the generalization of our algorithm with different obstacles and goals.

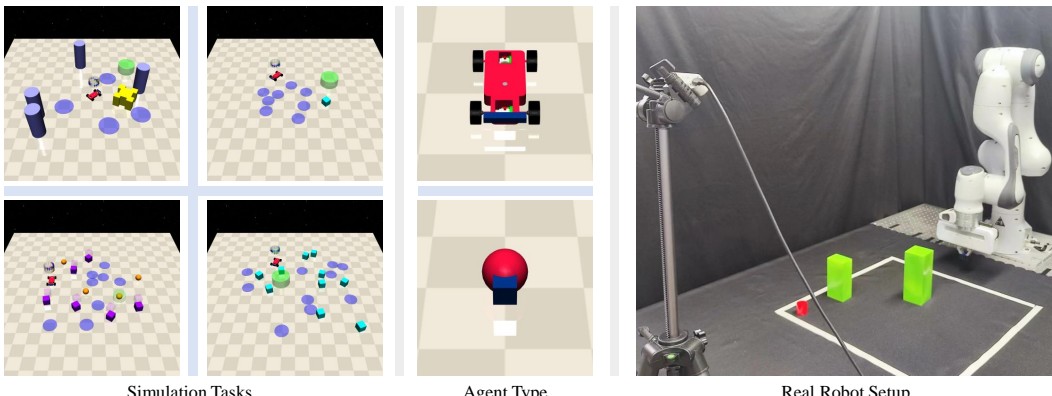

| Simulation Tasks | Agent Type | Real Robot Setup |

*Figure 7.* **Simulation and real-world environment.** Different tasks and agents in the simulation and real-world. (a) Simulation tasks: we consider four different tasks in the simulation: Push1, Goal1, Button1, Goal2. (from upper-left to lower-right) The preceding words indicate the task type and the following number represents the difficulty level. (b) Agent type: Car (upper) and Point (lower). (c) Real-robot setup: we use raw images as inputs and enable the robotic arm to complete obstacle avoidance tasks safely.

where it did not touch any obstacles but also did not successfully reach the target; 40 trajectories where it touched obstacles but successfully reached the target; 10 trajectories where it touched obstacles and did not successfully reach the target. Each trajectory contains 80 data points at different time steps. Each data point includes state: a 64×64 RGB image captured by the camera, a 6-dimensional pose of the robotic arm; action: a 6-dimensional variation recording the changes in the robotic arm's pose; cost: 1-dimensional cost index indicating whether the robotic arm touched an obstacle; reward: 1-dimensional reward index indicating whether the object reached the target object. The labeling methods for the cost and reward index are shown in Figure 8. The cost is assigned a value of 1 when the robotic arm collides with an obstacle, and 0 otherwise. Similarly, the reward is set to 1 when the object reaches the target and 0 in all other cases. We defined 3 placement points for target objects and 6 placement points for obstacles in the scenario. By restricting the placement positions of the target objects and obstacles, we ensure that the model can converge more efficiently on a limited dataset.

**Transfer Tasks**  To validate the generalization of the algorithm, we design a series of transfer experiments. As illustrated in the upper side of Figure 6, the original task involves 2 green rectangular obstacles and 1 red cylindrical target object. We design three different transfer tasks, depicted on the lower side of Figure 6. The tasks include altering the shape of the obstacles, changing the shape of the target object, and increasing the number of obstacles. These experiments aim to verify the algorithm's generalization in avoiding obstacles of various shapes and quantities, as well as in reaching target objects of different shapes.

## D.2  SIMULATION ENVIRONMENT

**Metrics**  In the simulation environment, we use three metrics to measure the performance:

- Reward $J_r$ is the average episodic sum of rewards.

$$J_r = \frac{1}{E} \sum_{i=1}^{E} \sum_{t=1}^{T_{ep}} r_{t,i},$$

- Cost $J_c$ is the average episodic sum of costs.

$$J_c = \frac{1}{E} \sum_{i=1}^{E} \sum_{t=1}^{T_{ep}} c_{t,i},$$

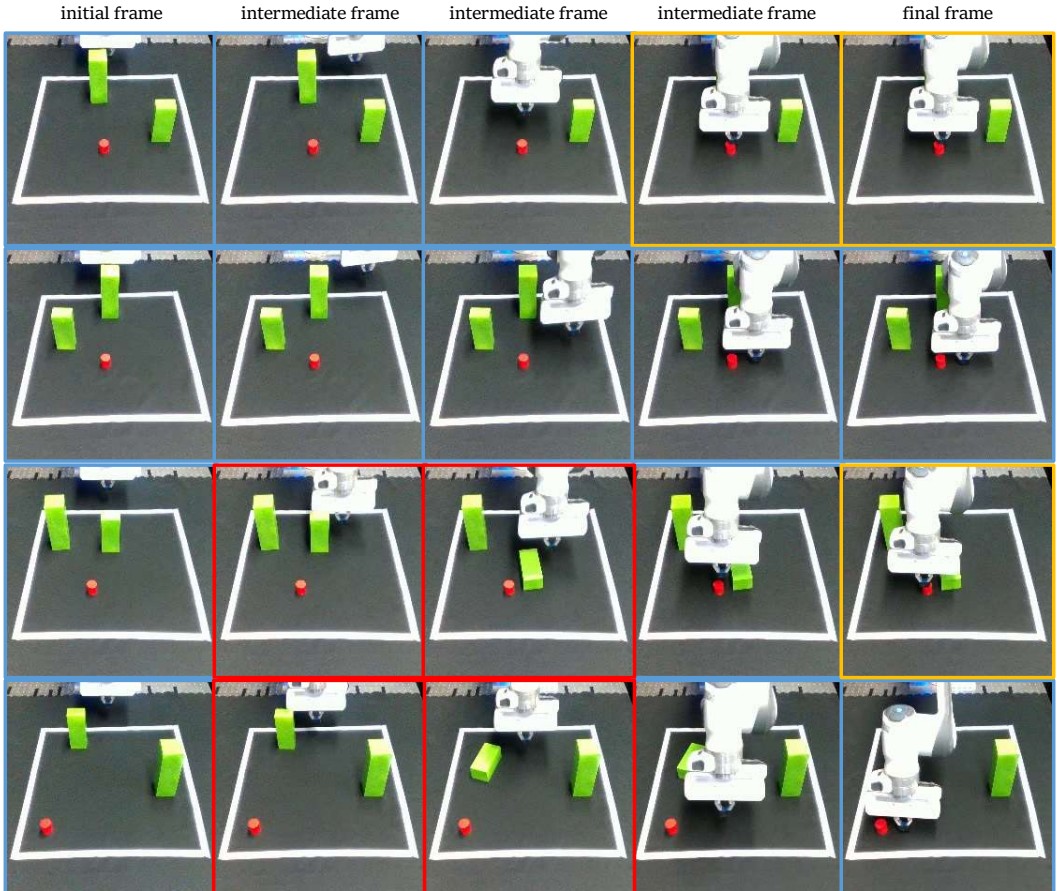

*Figure 8.* **Dataset collection.** Each row in the figure represents five frames of the robotic arm's motion captured from the same single trajectory. From the moment the robotic arm starts touching an obstacle to the moment the obstacle's posture stops changing, the cost label in the dataset is marked as 1, while in other cases it is marked as 0, as shown in the red-framed part in the image. When the robotic arm touches the target object, the reward label in the dataset is marked as 1, as shown in the yellow-framed part in the image.

- Cost Regret $\rho_c$ means the average cost over the entirety of training.

$$\rho_c = \frac{\sum_{t=1}^{T} c_t}{T},$$

where $E$ is the number of episodes, $T_{ep}$ is the timestep of episodes and $T$ is the total timestep.

**SafetyGoal** The agent's objective is to reach a goal while avoiding obstacles in its environment. Each time the agent successfully reaches the designated goal, the environment randomly generates a new one. The agent earns rewards for approaching or reaching the goal but incurs penalties when it encounters obstacles. These obstacles include immovable hazards and movable vases. The agent's observation state is represented by the images from its front and back. Two specific tasks are selected: the Point-Goal task and the Car-Goal task, which are performed by the Point and Car agents, respectively. The Point agent is a robot that operates on a 2D plane and is capable of rotating and moving both forward and backward. On the other hand, the Car agent, which is slightly more complex, features two independent parallel wheels and a freely rolling rear wheel.

**SafetyButton** The agent's goal is to navigate around both stationary and moving obstacles in the environment to press one of several target buttons. Similar to the goal task, the agent is rewarded for

approaching or pressing the target button. The Button task introduces dynamic obstacles that move quickly along set paths, making it more challenging than the Goal task due to the costs incurred from collisions with these moving obstacles.

**SafetyPush**    The agent's goal is to push an object to reach a desired goal while avoiding hazards and vases in the environment. It will get rewards when the object successfully pushes the yellow object to the goal. The agent does not fully control the object's movement, making the task more difficult to handle. And there are also some line-of-sight obstruction problems in vision-only situations.

**SafetyFading**    This environment is similar to the SafetyGoal, where the agent needs to reach a goal while avoiding obstacles. However, over time, the goal and obstacles gradually become undetectable. The agent needs to gather as much information about the environment as possible from the initial stage and form a memory of the goal and obstacle locations to complete the task according to its initial plan. This is undoubtedly more challenging than SafetyGoal and is even harder to accomplish for visual-only input agents.

### D.3    SIMULATION DATASET

We collect the simulation standard dataset from three policies: random policy, safe policy and unsafe policy. Each policy collected 200 trajectories for each task, which were then combined into the final dataset. This approach ensures that our dataset uniformly includes all types of data: low cost with low return, low cost with moderate return, high cost with low return, and high cost with high return. Thus, it can make a better trade-off in training different offline policies. The safe policy is trained by SafeDreamer (Huang et al., 2023) and the unsafe policy is trained by DreamerV3 (Hafner et al., 2023) for 1.5M steps on each task. The standard dataset distributions on five simulation tasks are illustrated in Figure 9. They show that unsafe policies tend to collect data with higher returns but significant cost uncertainty in these scenarios.

## E    DETAILS OF REAL-WORLD EXPERIMENTS

We illustrate the details of real-world experiments in Figure 10. The FOSP is first pretrained on an offline dataset for 0.5 million steps and deployed in the real world to execute the unseen tasks. In each task, we recorded the robot's trails before and after fine-tuning over 40 trials. As we can see, the robot's ability to safely complete unseen tasks has significantly improved after fine-tuning. It shows the strength of fine-tuning on different tasks.

We also compare our method with SafeDreamer on these tasks after fine-tuning. Both methods are pretrained for 500,000 steps and undergo 40 fine-tuning iterations. As depicted in Figure 11, SafeDreamer sometimes violates constraints even when the task is completed. In contrast, FOSP can learn with zero constraint violations and ultimately reach the goal.

## F    BASELINES

**DreamerV3**    DreamerV3 (Hafner et al., 2023) is a model-based reinforcement learning method that outperforms specialized methods across over 150 diverse tasks. It has a stable performance without adjusting hyperparameters in exploring farsighted policies from pixels and sparse rewards in an open world. However, it overlooks the safety considerations in the environment, which brings high costs in safety-critical tasks.

It uses RSSM as the world model, and the loss function is similar to equation 1 but lacks the cost head.

$$\mathcal{L}^{\text{model}}(\theta) = \mathbb{E}_{\tau \sim \mathcal{D}} \Big[ \sum_{t=1}^{T} -\ln p_\theta(\boldsymbol{x}_t \mid \boldsymbol{s}_t) - \ln p_\theta(\boldsymbol{r}_t \mid \boldsymbol{s}_t) + \mathbb{D}_{KL}[sg(q_\theta(\boldsymbol{z}_t|\boldsymbol{h}_t,\boldsymbol{x}_t))||p_\theta^{i_t}(\boldsymbol{z}_t|\boldsymbol{h}_t)] +$$
$$\beta \mathbb{D}_{KL}[q_\theta(\boldsymbol{z}_t|\boldsymbol{h}_t,\boldsymbol{x}_t)||sg(p_\theta^{i_t}(\boldsymbol{z}_t|\boldsymbol{h}_t))] \Big].$$

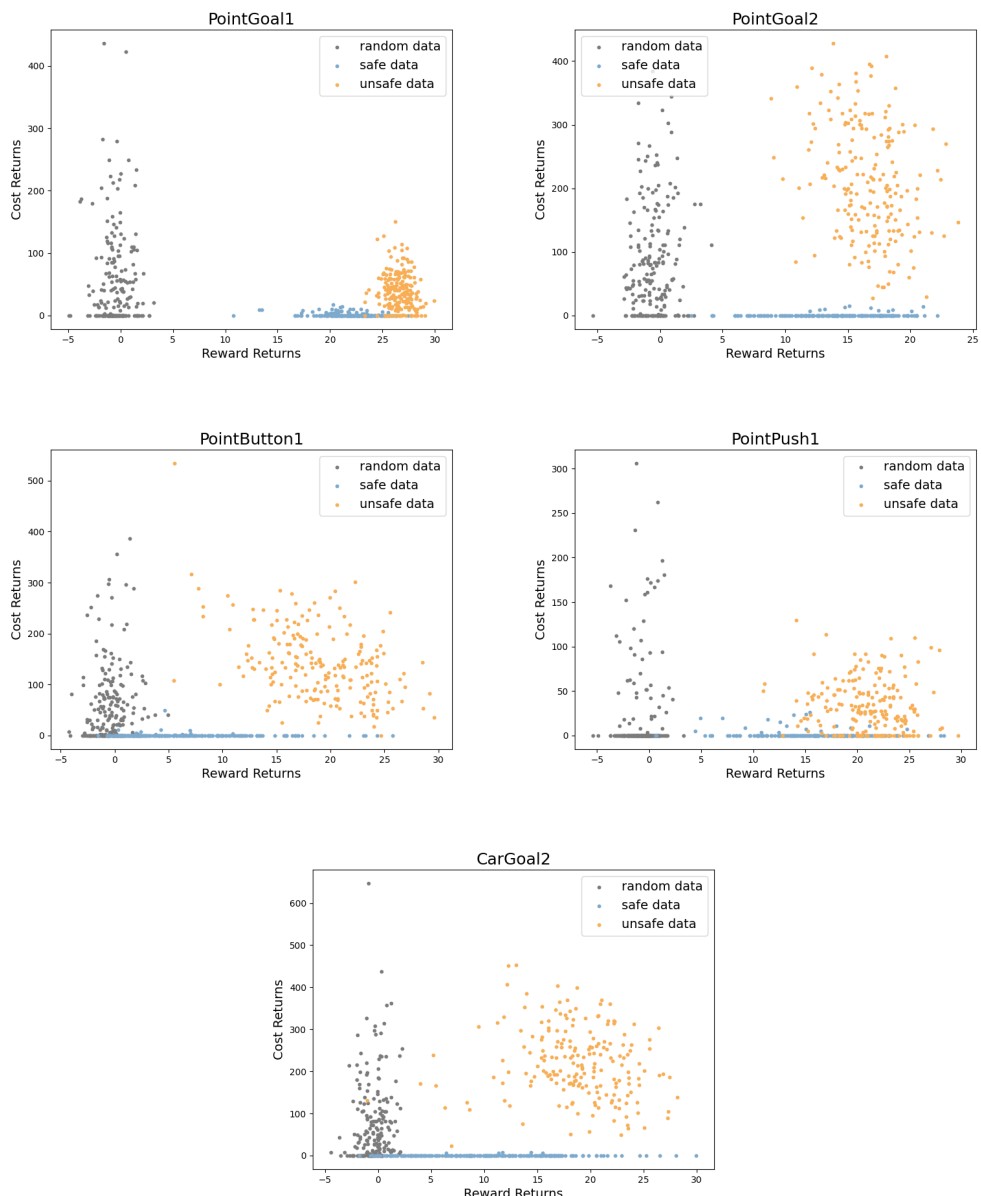

*Figure 9.* **The dataset distribution on five simulation tasks.** We plot the reward return and the cost return of every data trajectory. Gray dots represent random data, yellow dots represent unsafe data, and blue dots represent safe data.

And a simple actor-critic framework is used in its decision-making part.

$$\text{Actor: } \boldsymbol{a}_t \sim \pi_\psi(\boldsymbol{a}_t|\boldsymbol{s}_t) \quad \text{Critic: } V_\phi(R_t|\boldsymbol{s}_t) = \mathbb{E}\Big[\sum_t \gamma^t \boldsymbol{r}_t\Big]$$

$$\text{Actor loss: } L(\psi) = -\sum_t ((R_t - V_\phi(\boldsymbol{s}_t))/\max(1, S))\log \pi_\psi(\boldsymbol{a}_t|\boldsymbol{s}_t) + \eta H\Big[\pi_\psi(\boldsymbol{a}_t|\boldsymbol{s}_t)\Big]$$

$$\text{Critic loss: } L(\phi) = -\sum_t \log p_\phi(R_t|\boldsymbol{s}_t)$$

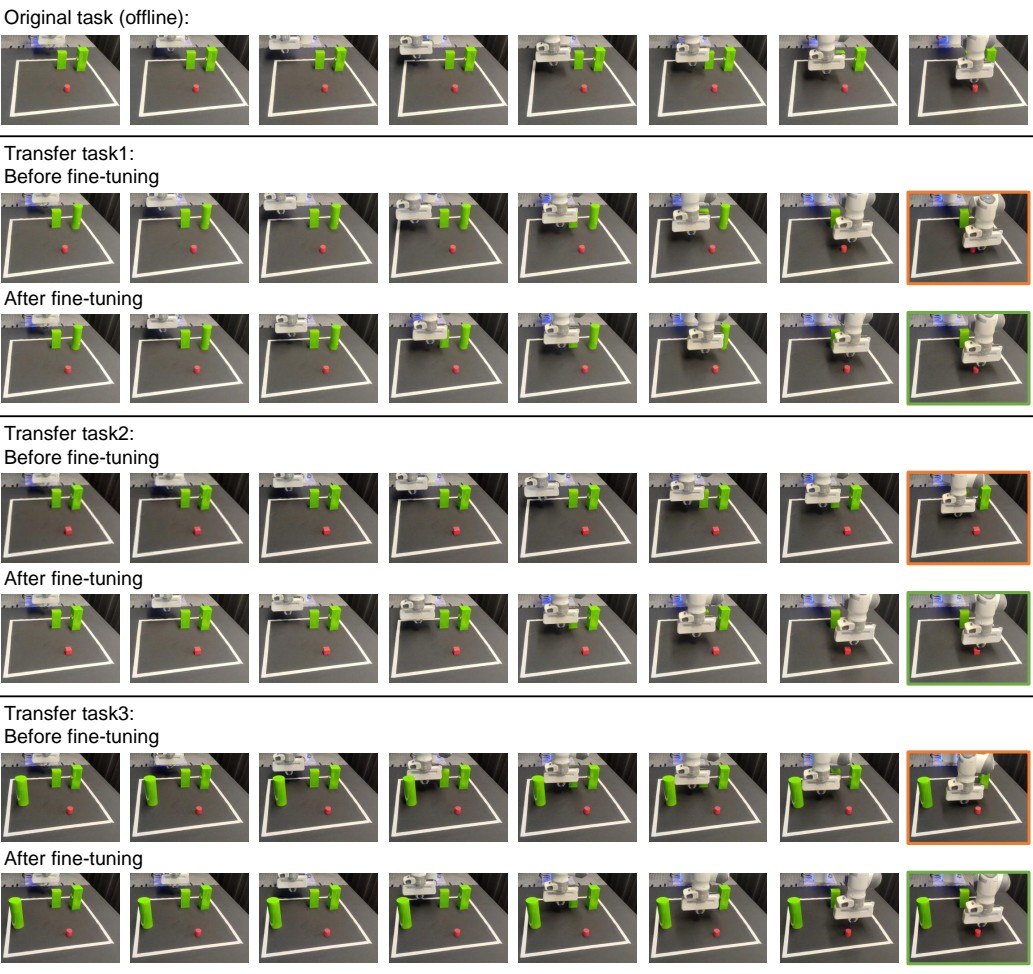

*Figure 10.* **Deployment on different real-world unseen tasks.** The trails from top to bottom show the original offline training task and three unseen fine-tuning tasks, with each fine-tuning task displaying both before and after fine-tuning trails. The green frames indicate successful task completion, while the orange frames represent failures.

In the offline pretraining phase, we removed the online planning component and used it only for online fine-tuning. To achieve robust performance in the offline phase, we train a latent dynamics model ensemble and use the uncertainty estimation approach (Yu et al., 2020):

$$\hat{\boldsymbol{r}}_\theta(\boldsymbol{s}_t, \boldsymbol{a}_t) = \boldsymbol{r}_\theta(\boldsymbol{s}) - \alpha \cdot \text{std}(\{\log(p_\theta^i(\boldsymbol{z}|\boldsymbol{h}))\}_{i=1}^N),$$

where $\boldsymbol{r}_\theta(\boldsymbol{s})$ is the reward predicted by the reward encoder.

**SafeDreamer** SafeDreamer (Huang et al., 2023) incorporates Lagrangian-based methods into world model planning processes and achieves nearly zero cost performance on various tasks. It performs well in high-dimensional vision-only input safety-critical tasks, surpassing the prior works (As et al., 2022; Hogewind et al., 2022) and balancing performance and safety. The framework uses the same world models based on DreamerV3 (Hafner et al., 2023). We use the BSRP-Lag version of the model, which utilizes the Lagrangian method in background safety-reward planning that avoids online planning. In the offline setting, we train it by providing the offline dataset and removing the parts involving interaction with the environment. This version of SafeDreamer also achieves the best performance of the three versions.

It adds a cost head into world models like our method and uses the same loss equation 1 in model training. It constructs a cost critic $V_\phi^c(R_t|\boldsymbol{s}_t)$ like $V_\phi(R_t|\boldsymbol{s}_t)$ and utilize Augmented Lagrangian

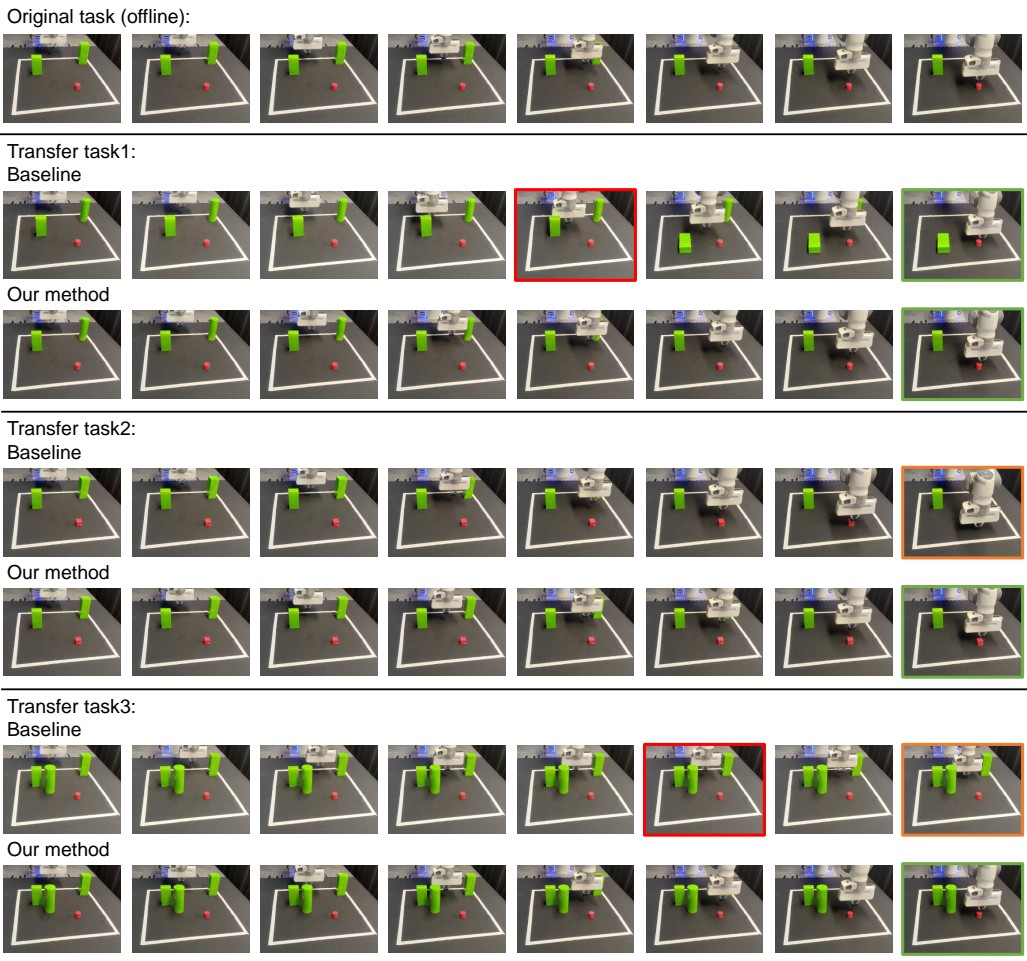

*Figure 11.* **Compare with baseline on different real-world unseen tasks.** In each task, we compare our method with the fine-tuned SafeDreamer (baseline). The green frames indicate successful goal reaching, the red frames indicate constraint violations and the orange frames represent failures.

method to update the actor:

$$\mathcal{L}(\theta) = -\sum_{t=1}^{T} R^\lambda(\boldsymbol{s}_t) + \eta \mathrm{H}\left[\pi_\theta\left(\boldsymbol{a}_t \mid \boldsymbol{s}_t\right)\right] - \Psi\left(C^\lambda(\boldsymbol{s}_t), \lambda_p^k, \mu^k\right),$$

$$\Psi\left(C^\lambda(\boldsymbol{s}_t), \lambda_p^k, \mu^k\right), \lambda_p^{k+1} = \begin{cases} \lambda_p^k \Delta + \frac{\mu^k}{4}\Delta^2, \lambda_p^k + \frac{\mu^k}{2}\Delta, & \text{if } \lambda_p^k + \frac{\mu^k}{2}\Delta \geqslant 0, \\ -\frac{\left(\lambda_p^k\right)^2}{\mu^k}, 0, & \text{otherwise,} \end{cases}$$

where $\delta = C^\lambda(\boldsymbol{s}_t) - b$.

**Recovery RL** Recovery RL (Thananjeyan et al., 2021) leverages offline data to learn the constraint violation zones before learning and uses two policies to separate the goal of enhancing performance and satisfying the constraints. It achieves nearly zero cost in uncertain environments where safety limits exploration. We use the model-based version of this baseline. First, we train cost Q-value by the following MSE loss:

$$\mathcal{L}(\phi) = (Q_\phi^c(\boldsymbol{s}_t, \boldsymbol{a}_t) - (\boldsymbol{c}_t + (1 - \boldsymbol{c}_t)\gamma_c \mathbb{E}_\pi[Q_\phi^c(\boldsymbol{s}_{t+1}, \boldsymbol{a}_{t+1})]))^2.$$

*Table 4.* **Hyperparameters for FOSP**

| Module | Name | Symbol | Value |
|---|---|---|---|
| World Model | Number of latent | $N_l$ | 48 |
| | Classes per latent | $C_l$ | 48 |
| | Batch size | $B$ | 64 |
| | Batch length | $T$ | 16 |
| | Learning rate | $l_{wm}$ | $10^{-4}$ |
| | Coefficient of KL-divergence | $\beta$ | 0.1 |
| | Generation horizon | $H$ | 15 |
| Augmented Lagrangian | Penalty term | $\nu$ | $5^{-9}$ |
| | Initial Penalty multiplier | $\mu^0$ | $1^{-6}$ |
| | Initial Lagrangian multiplier | $\lambda_p^0$ | 0.01 |
| Actor Critic | Discount horizon | $\gamma$ | 0.997 |
| | Reward lambda | $\lambda_r$ | 0.95 |
| | Cost lambda | $\lambda_c$ | 0.95 |
| | Expectile | $\kappa$ | 0.8 |
| | AWR temperature | $\beta_1, \beta_2$ | 10 |
| | REF discount | $\gamma_u$ | 0.99 |
| | PEX temperature | $\alpha$ | 10 |
| | Actor entropy regularize | $\eta$ | $3 \cdot 10^{-4}$ |
| | Learning rate | $l_{ac}$ | $3 \cdot 10^{-5}$ |
| | REF Learning rate | $l_r$ | $5 \cdot 10^{-5}$ |
| General | Number of MLP layers | $N_{\text{MLP}}$ | 5 |
| | Number of MLP layer units | $N_{\text{units}}$ | 512 |
| | Action repeat | $n_{\text{repeat}}$ | 4 |

Then we select actions from the safe set and the recovery set:

$$\boldsymbol{a}_t = \begin{cases} \boldsymbol{a}_t^{\pi_{\text{task}}}, & \text{if } (\boldsymbol{s}_t, \boldsymbol{a}_t) \in \{(\boldsymbol{s}, \boldsymbol{a}) \in \mathcal{S} \times \mathcal{A} : Q_\phi^c(\boldsymbol{s}, \boldsymbol{a}) \leqslant \epsilon_c\}, \\ \boldsymbol{a}_t^{\pi_{\text{recovery}}}, & \text{if } (\boldsymbol{s}_t, \boldsymbol{a}_t) \in \{(\boldsymbol{s}, \boldsymbol{a}) \in \mathcal{S} \times \mathcal{A} : Q_\phi^c(\boldsymbol{s}, \boldsymbol{a}) > \epsilon_c\}, \end{cases}$$

where $\epsilon_c$ is a threshold. We follow the model predictive control (MPC) as a learned dynamic model $f_\theta$ and use a VAE-based model to capture the high-dimensional information. And we utilize SAC (Haarnoja et al., 2018) to learn $\pi_{\text{task}}$.

**PPO-Lagrangian** PPO-Lagrangian uses the objective of clipped PPO (Schulman et al., 2017) to optimize:

$$\mathcal{L}(\theta)_{ppo} = \min(\frac{\pi_\theta(\boldsymbol{a}|\boldsymbol{s})}{\pi_{\theta_k}(\boldsymbol{a}|\boldsymbol{s})} A_r^{\pi_{\theta_k}}(\boldsymbol{s}, \boldsymbol{a}), \text{clip}(\frac{\pi_\theta(\boldsymbol{a}|\boldsymbol{s})}{\pi_{\theta_k}(\boldsymbol{a}|\boldsymbol{s})}, 1 - \epsilon_{\text{clip}}, 1 + \epsilon_{\text{clip}}) A_r^{\pi_{\theta_k}}(\boldsymbol{s}, \boldsymbol{a})),$$

We use the PID Lagrangian (Stooke et al., 2020) method and obtain the loss of the PPO-Lagrangian:

$$\mathcal{L}(\theta)_{ppol} = \frac{1}{1 + \lambda}(\mathcal{L}(\theta)_{ppo} - \lambda A_c^{\pi_{\theta_k}}(\boldsymbol{s}, \boldsymbol{a})).$$

**CPO** CPO (Achiam et al., 2017) uses a local policy search combined with trust region recovery to ensure that single-step policy updates follow a direction that does not violate the constraints. It introduces this form to optimize the problem:

$$\theta^* = \theta_k - \sqrt{\frac{2\delta}{b^T H^{-1} b}} H^{-1} b,$$

where H is the e Hessian of KL-divergence.

## G HYPERPARAMETERS

The experiments for FOSP were conducted in a Python 3.10 environment with JAX 0.4.26. Our setup included CUDA version 12.1, running on Ubuntu 20.04. The hardware used comprised four

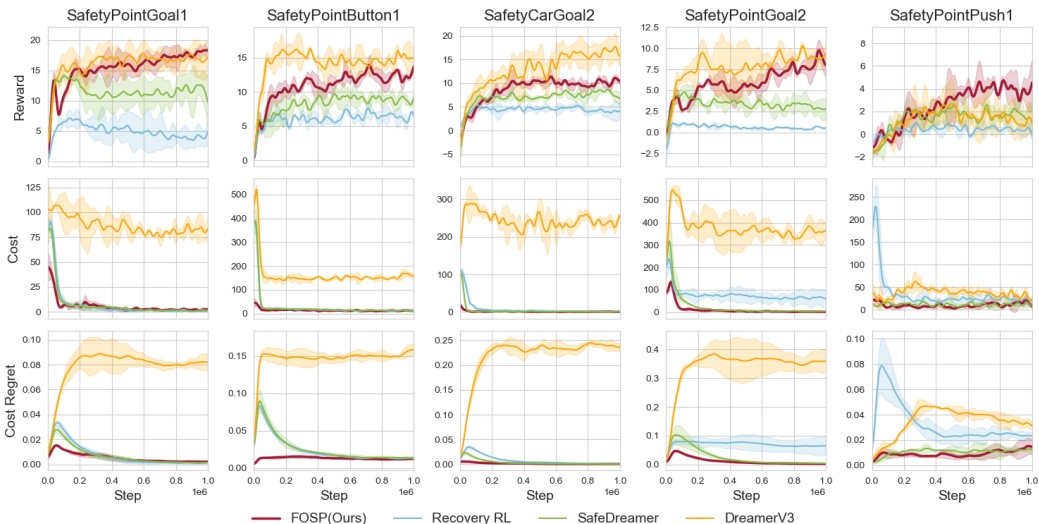

*Figure 12.* **Offline experimental results.** Comparing FOSP to baselines across five image-based safety tasks. The results for all three algorithms are obtained after training for 1 million steps. Reward: averaged episode reward return. Cost: averaged episode cost return. Cost Regret: averaged cost value throughout the training phase. As we can see, FOSP can maintain safety and achieve better performance during offline training.

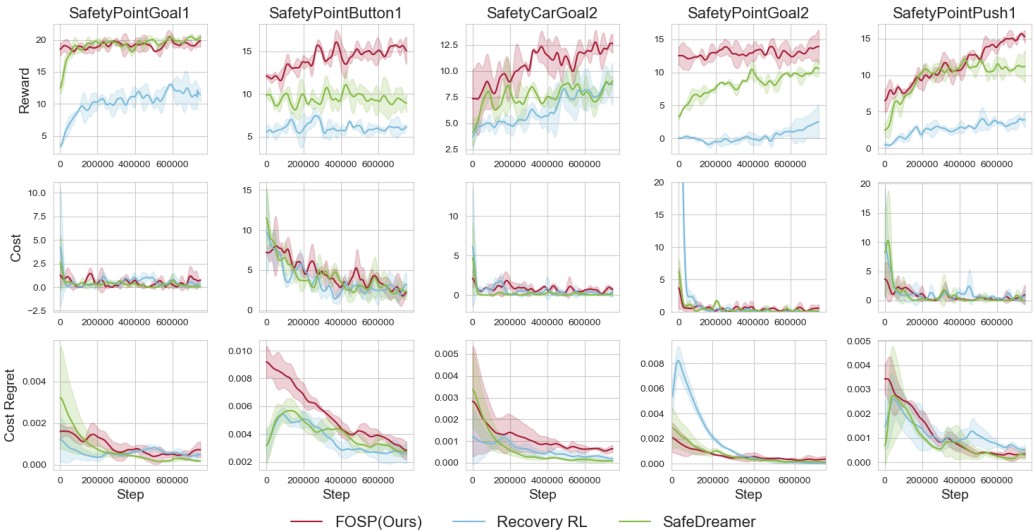

*Figure 13.* **Online experimental results without DreamerV3.** These results are the same as Figure 2. We omit the curves of some baselines to clearly illustrate the results of other main methods.

GeForce RTX 4090 GPUs and an Intel(R) Xeon(R) Platinum 8358P CPU @ 2.60GHz. And the experiments' hyperparameters setting is shown in Table 4.

# H    ADDITIONAL EXPERIMENTAL RESULTS

## H.1    OFFLINE TRAINING RESULTS

We present the training curves of FOSP during offline pretrain in simulation experiments in Figure 12. The results show that with minimal fine-tuning, FOSP achieves better performance, outperform-

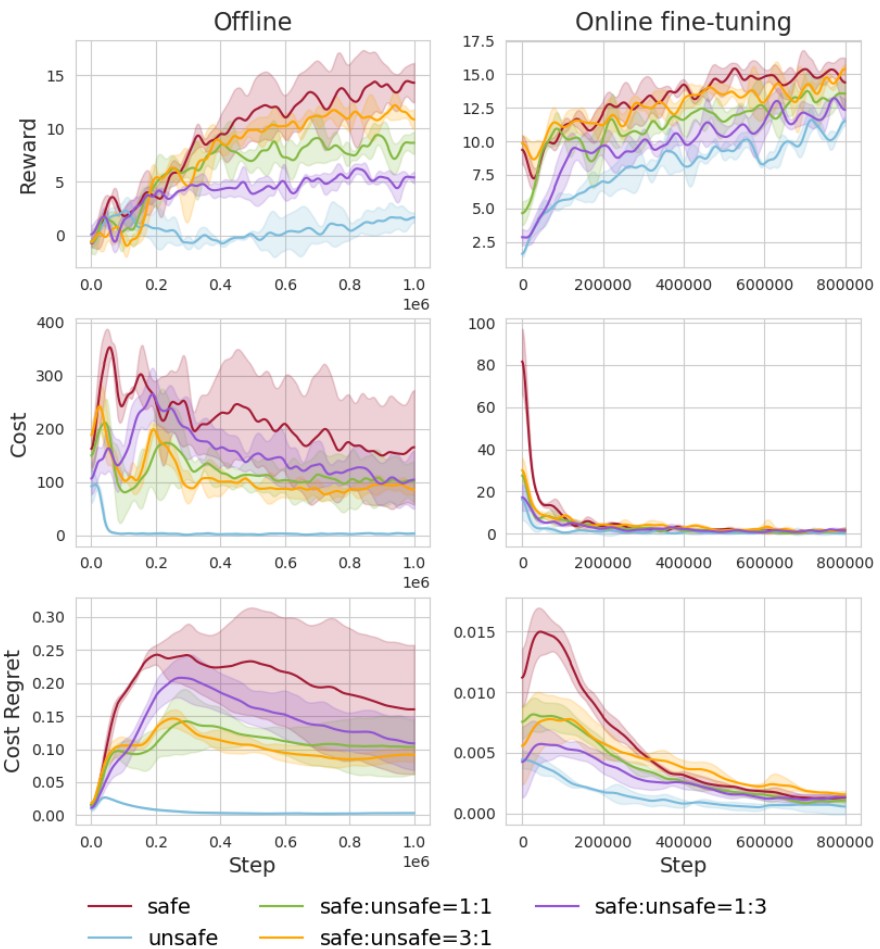

*Figure 14.* **Results on dataset ablation.** We conducted experiments with five different dataset ratios, performing an ablation study on the proportion of safe and unsafe data. Each model was trained offline for 1 million steps and fine-tuned online for 0.8 million steps.

ing SafeDreamer. Although it does not match the performance of DreamerV3, FOSP achieves nearly zero cost of the agent throughout the entire process. Note that all methods struggle to get higher rewards in SafetyPointPush because the line-of-sight obstructions necessitate online exploration.

## H.2   DATASET ABLATION RESULTS

The detailed training curves of dataset coverage ablation are shown in Figure 14. They all trained on the SafetyPointGoal2 for 1 million steps during the offline stage and 0.8 million steps for online fine-tuning. The dataset size is limited to 600 trajectories. As we can see, due to the characteristics of model-based safe RL, a high-cost, high-reward policy is learned from a purely safe dataset, while both cost and reward are low from a purely unsafe dataset. This is because the uneven distribution of the dataset leads to biases in the training of the dynamics model. Since model-based reinforcement learning algorithms rely heavily on model-generated rollouts for training, this bias can significantly impact the final decision-making. If the dataset only contains safe data, the dynamics model trained through supervised learning will mistakenly assume that the agent will always incur a cost of 0 in any state, leading the agent to ignore dangerous areas (failing to learn the cost critic). Conversely, suppose the dataset only contains unsafe data. In that case, the model will generate a large number of unsafe states that hinder the agent from completing the task, ultimately causing the agent to lose the ability to accomplish the task (overlearn the cost critic).

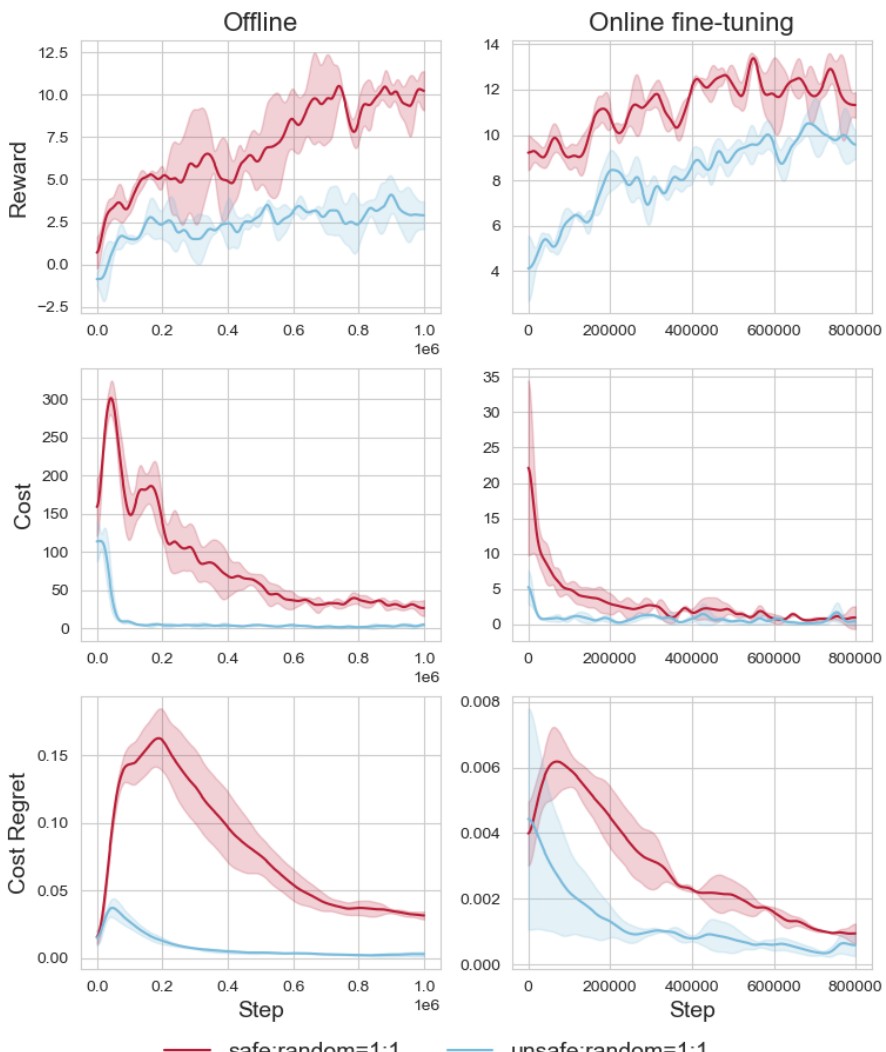

*Figure 15.* **Dataset ablation with random data.** We designed a series of comparative experiments involving mixtures of safe/unsafe data and random data. Each model was trained offline for 1 million steps and fine-tuned online for 0.8 million steps.

The dataset coverage further influences online performance. Due to offline training results, the policy pertrained on a safe dataset will have higher rewards with aggressive behaviors while the unsafe one will be more conservative on the initial stage of fine-tuning. It also affects the final performances. As a result, we adopt a compromise approach to train the policy (i.e. green curve), which achieves relatively higher rewards while maintaining zero cost.

We also illustrate the safe-random and unsafe-random mixed experiments to investigate the impact of random data in Figure 15. Compared to experiments on purely safe or purely unsafe datasets (Figure 14), adding random data can help alleviate errors in learning the world model to some extent. As shown in the training curves, experiments mixing safe data with random data resulted in lower costs during offline training than purely safe experiments. And mixing unsafe data with random data enhanced the model's exploration, getting relatively higher rewards. Consequently, we need to add some random data to the dataset to get better performance.

The dataset size ablation studies are shown in Figure 16. The results align with general expectations. The policies trained on larger datasets outperform those trained on smaller datasets. However, as the amount of data increases, the improvements become less significant.

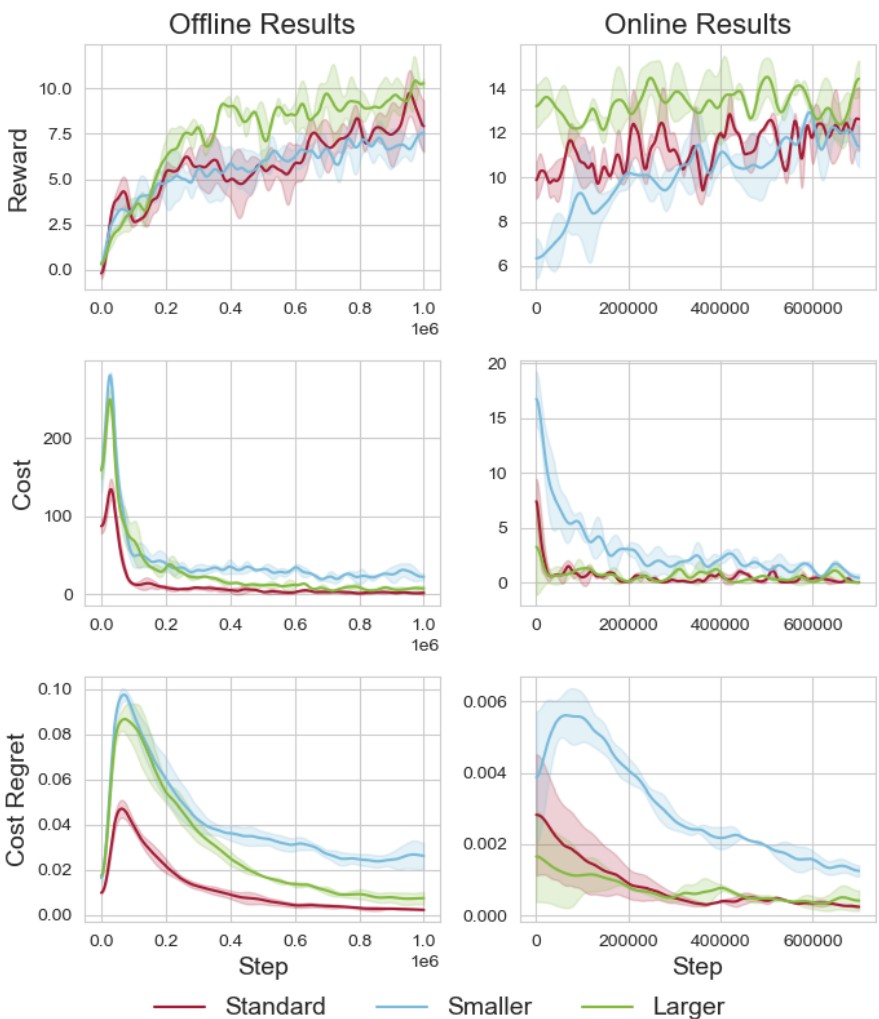

*Figure 16.* **Dataset ablation with different sizes.** We illustrate the details of dataset size ablation studies. "Standard" represents standard size, which contains 600 trajectories. "Larger" and "Smaller" denote the larger one containing 900 trajectories and the smaller one containing 300 trajectories. Each model was trained offline for 1 million steps and fine-tuned online for 0.7 million steps.

### H.3 SAFE GENERALIZATION RESULTS

As depicted in Figure 17, we devise more simulation experiments on safe generalization tasks. The upper one shows the performances during the offline phase. The following two figures show their generalization performance on the more challenging tasks, *SafetyFadingEasy1* and *SafetyFadingHard1*. During the fine-tuning process, the target will gradually disappear and the agent should quickly find a path to reach the goal. The results demonstrate that FOSP can get higher rewards and lower costs even though it is deployed on tasks different from offline pretraining. Compared to SafeDreamer, it is significantly important that it can maintain near zero constraint violations during fine-tuning on new tasks.

We test the specific average reward and cost for the transfer task from *SafetyPointGoal1* to *SafetyPointGoal2* and present the results in Figure 18. Although FOSP can outperform SafeDreamer, it may also face some costs caused by novel constraints. As shown in Figure 18, the initial cost in the new environment is slightly higher than the original but reduces quickly as fine-tuning progresses. It indicates that the safe policy expansion mechanism helps the model adapt to new safety

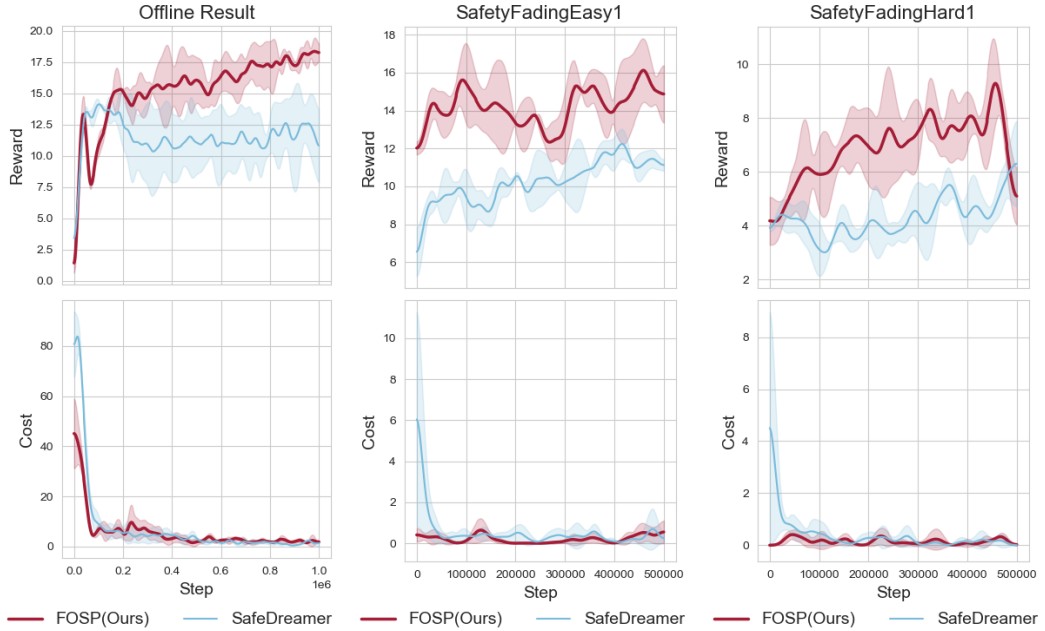

*Figure 17.* **More experiments on safe generalization tasks.** We evaluate our method on different safe generalization tasks. The upper one illustrates the offline pretrain performances on Safety-PointGoal1. We then fine-tune the model on SafetyFadingEasy1 and SafetyFadingHard1. FOSP can consistently outperform the baseline while ensuring safe fine-tuning.

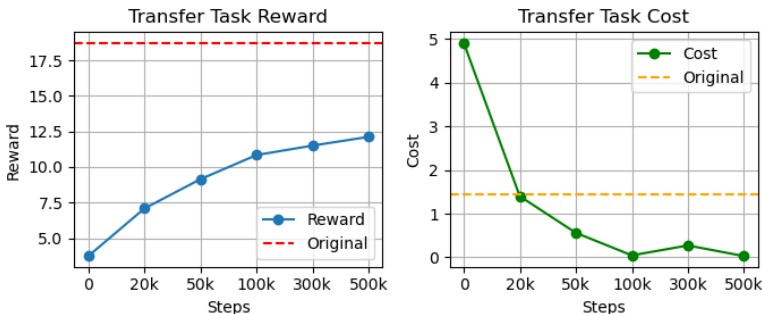

*Figure 18.* **Results on average reward and cost for transfer task.** The results are evaluations on SafetyPointGoal1 to SafetyPointGoal2 task. The x-axis represents the number of fine-tuning steps. The original means the model's performance at the end of offline pretraining in the original environment.

constraints quickly. The fine-tuning process can further improve its safety performance, promoting better generalization.

### H.4 COMPARISONS BETWEEN ONLINE TRAINING AND FINE-TUNING

To further validate the effectiveness of the fine-tuning process, we compare the results between FOSP directly trained online and FOSP with fine-tuning. We illustrate the results for different numbers of steps in the Table 5. The results denote that the online version FOSP does not perform as well as the fine-tuning method, even after training for 1M steps. Meanwhile, direct online training faces challenges in requiring a large number of training steps to converge. In contrast, our framework allows the agent to leverage offline knowledge to enhance online performance effectively with relatively short fine-tuning steps. The results also suggest that the policy can be continually improved during online fine-tuning.

*Table 5.* **Online fine-tuning and direct online results.** Reward return and cost return of FOSP in offline-to-online fine-tuning (1M steps for offline pretraining) and direct online training. We report the mean value of 5 independent runs with different seeds.

| Task | Metric | Online fine-tuning | | | | | | Direct online | |
|------|--------|--------|----------|----------|-----------|-----------|-----------|------------|-------------|
| | | 0 step | 20k steps | 50k steps | 100k steps | 300k steps | 500k steps | 500k steps | 1000k steps |
| PointGoal1 | Reward ↑ | 18.723 | 18.692 | 19.581 | 20.063 | 19.649 | 21.517 | 14.749 | 18.586 |
| | Cost ↓ | 1.451 | 3.4 | 2.719 | 0.328 | 0.268 | 0.2 | 2.66 | 1.08 |
| PointButton1 | Reward ↑ | 14.65 | 13.415 | 13.569 | 15.306 | 17.86 | 18.102 | 10.57 | 12.81 |
| | Cost ↓ | 9.622 | 8.639 | 6.664 | 4.8661 | 4.256 | 2.188 | 6.88 | 2.76 |
| PointPush1 | Reward ↑ | 4.092 | 7.253 | 7.916 | 9.148 | 10.373 | 13.281 | 1.397 | 4.181 |
| | Cost ↓ | 18.117 | 1.232 | 1.497 | 1.089 | 0.507 | 0.157 | 2.49 | 2.05 |
| PointGoal2 | Reward ↑ | 8.1 | 10.31 | 10.157 | 12.446 | 13.034 | 13.556 | 9.471 | 10.717 |
| | Cost ↓ | 7.556 | 1.486 | 0.427 | 0.18 | 0.214 | 0.234 | 4.779 | 3.089 |
| CarGoal2 | Reward ↑ | 10.146 | 9.063 | 9.816 | 10.275 | 12.378 | 14.512 | 8.749 | 10.144 |
| | Cost ↓ | 1.618 | 1.063 | 0.385 | 0.394 | 0.18 | 0.071 | 2.12 | 1.21 |

## H.5 MORE VISUAL CHANGES ON TRANSFER TASK

To evaluate the performance of our method under significant visual variations, the model pre-trained on *SafetyPointGoal1* is fine-tuned in the *SafetyPointBuildingGoal1* environment, with the results presented in Figure 19. The new environment has the same task as the original one but the visual inputs are different. The agent struggles to finish the task since it fails to recognize the hazardous areas and goals in the new environment. Plus, the inaccurate visual demonstrations in the offline dataset significantly impact the performance during online fine-tuning.

Based on the simulation results, we can easily know that FOSP can not deal with significant visual changes in the real world. Swapping the colors of obstacles and goals could be a straightforward setting (i.e., making obstacles red and goals green). However, some prior work has demonstrated that robots are sensitive to the color (Feng et al., 2023) and the robot is highly likely to mistakenly identify the obstacles to the goal. Additionally, it violates our original intention of using colors to distinguish obstacles and goals. Hence, the robot will fail in this setting.

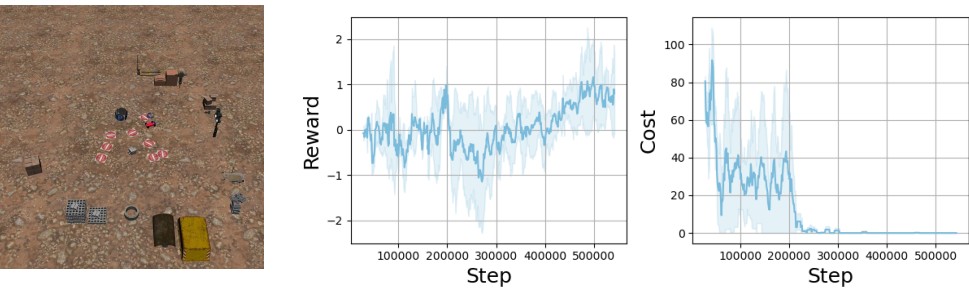

*Figure 19.* **Fine-tuning on SafetyPointBuildingGoal1.** The model pretrained for 1M steps on SafetyPointGoal1 is fine-tuned on SafetyPointBuildingGoal1. While the task objectives remain the same between these two environments, the agent's visual inputs are entirely different. In Safety-PointBuildingGoal1, the agent must avoid red-marked areas and reach the parking area (P). These hazardous areas and goals are totally different from the original ones.

## H.6 EXPERIMENTS ON RACE

We utilize a more realistic environment, Race from Ji et al. (2023), where the agent receives $64 \times 64 \times 3$ image inputs, as shown in Figure 20. An increase in environmental complexity better highlights the reliability of the algorithm. We employ Level 2 of the environment, which requires the agent to reach the goal position from a distant starting point while ensuring it avoids straying into the grass and prevents collisions with roadside objects. The offline dataset collected by standard procedures is a mixture of safe, unsafe and random data.

As illustrated in Figure 21, FOSP has superior performance and lower constraint violations during the offline training phase. In the online fine-tuning stage, it further optimizes safety performance

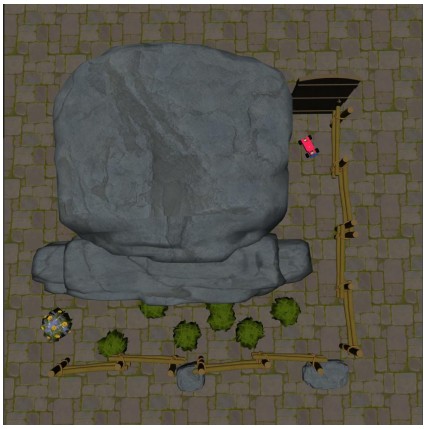 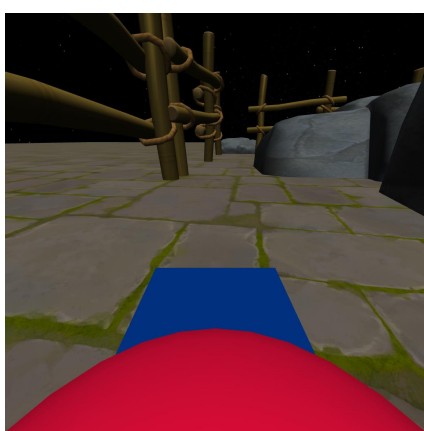

*Figure 20.* **Race environment.** The left subfigure shows the panoramic picture of the environment. The right subfigure is the first perspective of the agent.

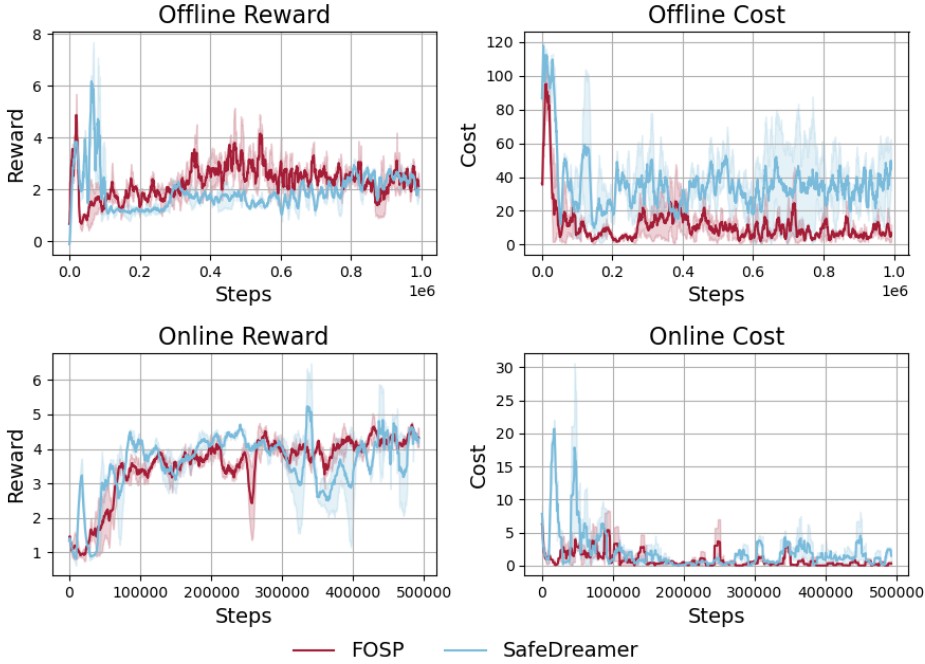

*Figure 21.* **Race experiments results.** We compare FOSP with SafeDreamer on Race2. Each model was trained offline for 1 million steps and online for 0.5 million steps. FOSP has comparable rewards but lower costs than SafeDreamer.

while ensuring a stable improvement in task rewards. Conversely, SafeDreamer exhibits higher constraint violations during the offline phase, leading to unsafe behaviors at the beginning of the online fine-tuning. Additionally, its safety design limitations make it challenging to achieve nearly zero constraint violations.

## I  LIMITATION AND FUTURE WORK

### I.1  LIMITATIONS

**Safe constraints**   Despite achieving good results in many simulated environment experiments, the proposed framework still has some safety concerns. First, it inevitably requires a balanced sampled and sufficiently large dataset for offline policy pertraining, which can be undesirable for safety-critical applications. An uneven dataset, such as one lacking unsafe data, or an insufficiently large dataset, can lead to deviations in model learning, thereby compromising its safety performance and generalization. Second, it may exceed the feasible region from time to time if the cost distribution has a long tail since CMDP only requires the policy to satisfy the expectation cost constraint. It might be considered that incorporating risk-constrained methods (Chow et al., 2018) can solve problems with long tail cost distributions.

**Real-world experiments**   In experiments with real robots, we find that the model performs poorly in tasks with occluded vision. Our test results remain unstable and are sensitive to learning rate adjustments. This indicates the need for significant effort in tuning the model's hyperparameters to achieve better performance. In addition, SafeReach task is quite preliminary. It is difficult to apply FOSP to more realistic scenarios like real-time control or dynamic obstacles as it lacks certain predictive capabilities. The model also fails to handle novel visual observations beyond scene re-configurations during fine-tuning because it struggles to interpret the meaning of different objects in new environments.

### I.2  FUTURE WORKS

**More complex real-world tasks**   To further decrease the violations, the safety component of our method can be improved during the offline-online stage. The SafeReach task is a preliminary experiment in the real world and it is possible to expand our method to general robot tasks such as grasping, pulling, and pushing. For the tasks with novel visual observations, leveraging semantic information into the inputs is a promising solution, which can help the agent attach meaningful interpretations to raw sensory inputs.

**Improvement in real-world scenarios**   The real-world task performance of our method can be further improved. To address the issue of inaccurate position determination from a single perspective and line of sight occlusion, the multi-view RL shows its efficiency in catching better features from images. Increasing the dataset capacity and uniforming the dataset distribution can probably enhance the robustness of the performance and expand it to more complex scenarios.

**Safe sim2real**   FOSP also shows potential in the field of safe sim-to-real transfer. Due to the limitations of simulators in fully replicating real-world environments, robots may encounter safety challenges when deploying algorithms in the real world. Thus, it is important to fine-tune it in the real world with safety considerations. Meanwhile, the use of world models will help speed up the fine-tuning process.

