# OpenReview forum: "FOSP: Fine-tuning Offline Safe Policy through World Models"
_ICLR.cc/2025/Conference — ICLR 2025 Poster_

### Official Review · Reviewer_UhwZ · 2024-11-04

**Soundness:** 3
**Presentation:** 2
**Contribution:** 3
**Rating:** 8
**Confidence:** 3

**Summary:**

This paper studies the problem of offline-to-online finetuning of learned policies with safety constraints. This is, to my knowledge, the first work to study policy safety in this setting; prior work either focus on either offline or online RL exclusively. The key contribution of this paper is an algorithm, FOSP, that builds upon model-based RL algorithm Dreamer to enable fast and safe offline-to-online finetuning on tasks with visual observations, both in simulation and on a real robot. The algorithm works by first pretraining a world model on an offline dataset (using the in-sample trick from IQL), and then finetuning the model during the online phase using safe policy expansion to minimize safety violations. Experimental results indicate that FOSP is effective at reducing constraint violations while maintaining reasonably good task performance (i.e. rewards) compared to prior methods for safe RL. Vanilla Dreamer without regards for safety achieves consistently high reward (more than any other approach) but has significant safety violation.

**Strengths:**

- The paper studies an interesting problem (offline-to-online safe RL) that to my knowledge has not been explored much (if at all) by prior work. I appreciate the substantial background and derivation of approach which will help readers appreciate the technical contributions more. I believe that there is adequate discussion of related work.
- The paper is generally well written and easy to follow. The paper has minor grammatical errors but they do not detract significantly from my understanding.
- The resulting algorithm appears to rather practical, and can be applied to tasks with visual observations in both simulation and on a real robot. A significant amount of prior work in safe RL does not consider visual observations nor real robot experiments.
- Baseline methods appear to be quite strong. I appreciate including vanilla Dreamer for comparison (it can be considered an ~upper bound in terms of task performance with no regard to safety), as well as SafeDreamer as a recent and highly related method.

**Weaknesses:**

- The considered tasks are fairly toy. It would be useful to know how the method performs on more realistic problem settings. I believe that the real-world task is a nice step in that direction, but being simply a reaching task it does not have any of the additional complexity that usually comes with real-world tasks: object interaction, real-time control, dynamic tasks (environment might change even with an all-zero action step). Especially the latter two can have significant safety consequences.
- A substantial limitation of the method appears to be the lack of control or predictability in how safe the model really is during finetuning / at test-time when presented with OOD data, e.g. novel visual observations beyond scene re-configurations. I did not find any significant discussion of this in the paper, so it would be useful to clarify (verbally or with empirical evidence) when the proposed method can be expected to succeed / fail in a transfer setting; see my question below for more context.

**Questions:**

I would appreciate if the authors can address (in whichever way they deem appropriate) my comments in the "weaknesses" section above, as well the following questions:

- How would the proposed method behave in a transfer task setting where the change is more visual in nature? E.g. the target object is now green and the obstacles are now red. I suspect that such novel settings would trick the method into substantial safety violations. It would be informative to either include experimental results or a written discussion on the types of transfer settings in which the method can be expected to maintain low safety violations.
- Do the authors plan to release code that reproduces their main experimental results? I did not find any mention of that in the paper.

---

> ### Author Response · Authors · 2024-11-20
>
> We are very grateful to the reviewer for pointing out some of the weaknesses it considers. We are very willing to respond to each of the weaknesses and also answer the reviewer's questions one by one.
>
> **Weaknesses:**
>
> **1. The tasks are relatively simple**
>
> We appreciate the reviewer for pointing out this limitation. We agree with the simplicity of the tasks. And the reasons why we devise these tasks are shown as follows:
> 1) The simulation is a benchmark, which can be used to evaluate our method’s effectiveness.
> 2) The real-world task is quite easy because it is a preliminary attempt. Experiments need to progress from simple to complex; if simple tasks cannot be accomplished, it is almost impossible to succeed in more challenging ones. Therefore, we first conducted SafeReach to evaluate the performance in real-world scenarios.
> However, end-to-end visual safe reinforcement learning still has a long way to go before being directly applicable to real-world tasks. There exists some work that uses vision-based RL for real-world tasks by sim2real [1] or combining with IL [2]. Conversely, their methods do not involve constraints in the task, which makes training easier. In addition, they incorporate many technical skills that make it work. To the best of our knowledge, our method is the first to apply safe RL to real-world safe generalization tasks and we have demonstrated its effectiveness and practicality (by these simple tasks). The proposed framework can be a contribution to the community so that subsequent researchers can study more complex scenarios.
>
> Actually, the real-time control or dynamic tasks the reviewer mentioned seem more like works from a different area. They are not the core issue of our research so our method cannot handle these tasks. Since these tasks need the agent to predict the environment, they are more about how to predict what might happen next based on visual observations, such as [3] using MPC to predict. The design of the network (predictor) is more responsible for these tasks rather than the RL controller. So, we think the modification of the forecast will be more effective in improving dynamic tasks’ performance and it is orthogonal to our method, which means the modification can be applied to our framework.
>
> We also provide some more realistic simulation tasks in the updated version (**Appendix H.6**) to further illustrate our method’s effectiveness.
>
> **2. The lack of control or predictability for novel visual observations beyond scene re-configurations**
>
> It seems difficult for us to deal with completely novel environments. Even though our method has generalization ability, it is difficult to handle completely different environments. We have tried it in simulation tasks. After pretraining on SafetyPointGoal1, we transferred it to SafetyPointBuildingGoal1, which is a totally different environment from the former one but the task is the same (the environment can be found at https://safety-gymnasium.readthedocs.io/en/latest/environments/safe_vision/building_goal.html). The results are shown as follows (fine-tuning 50k steps):
>
> |Reward|Cost|
> |----|----|
> |0.787|0.21|
>
> It is clear that vision-based safe RL cannot work in totally different environments during fine-tuning.
>
> We want to specifically answer it in the question part and we have added the simulation experiment in **Appendix H.5**.
>
> **Questions:**
>
> **1. Transfer task changes are more visual in nature.**
>
> The above simulation experiment results confirm that our method struggles with significant visual changes since our method is based on vision-only inputs. In real-world experiments, we aim to use colors to differentiate obstacles and goals to ensure the model can perceive obstacles and goals through visual inputs. Therefore, it is impossible to maintain performance when we swap the colors of obstacles and goals. As robot needs to distinguish obstacles and goals by their colors, they might mistakenly identify obstacles as goals, leading to direct collisions. It will not understand the constraints settings as we intended. It is highly likely to trick the method into substantial safety violations because it violates our original intention of using colors. A promising solution is adding semantic information to the inputs, telling the agent which color object is the obstacle and which color object is the target.
>
> We have added the discussion to **Limitation and Future Works**.

---

> ### Author Response · Authors · 2024-11-20
>
> **2. Release the code**
>
> Yes, we have already released our code in the **Supplementary Material** at the time of initial submission, though it is a bit messy and needs to be organized. Since our datasets are too large to be released, users should collect data by themselves, and then train the FOSP.
> Here is a guidance to collect the dataset:
> -    First, create a new folder “dataset”.
> -    Replace agent.py with agent_sd.py and behavior.py with behavior_sd.py. Run “python  FOSP/train.py --configs safedreamer --method safedreamer --run.script collect_dataset --run.from_checkpoint /xxx/checkpoint.ckpt --task safetygym_SafetyPointGoal1-v0 --jax.logical_gpus 0 --run.steps 50000” to collect safe data (unsafe data: replace safedreamer to dreamerv3). To find the checkpoint, users should download it from Hugging Face (from SafeDreamer [4]) https://huggingface.co/Weidong-Huang/SafeDreamer/tree/main.
> -    Find the npz files and add them into ./dataset/. Add the path to ./embodied/replay/saver.py: self.load_dir. Then follow the Readme.md to train the FOSP.
>
> We will then organize it with the datasets and release it on github.
>
> References:
>
> [1] Lei, Kun, et al. "Uni-o4: Unifying online and offline deep reinforcement learning with multi-step on-policy optimization." arXiv preprint arXiv:2311.03351 (2023).
>
> [2] Haldar, Siddhant, et al. "Teach a robot to fish: Versatile imitation from one minute of demonstrations." arXiv preprint arXiv:2303.01497 (2023).
>
> [3] Bing, Zhenshan, et al. "Safety Guaranteed Manipulation Based on Reinforcement Learning Planner and Model Predictive Control Actor." arXiv preprint arXiv:2304.09119 (2023).
>
> [4] Huang, Weidong, et al. "Safe dreamerv3: Safe reinforcement learning with world models." arXiv preprint arXiv:2307.07176 (2023).

---

> > ### Comment · Reviewer_UhwZ · 2024-11-22
> > **Thank you**
> >
> > Thank you for responding to my comments in great detail.
> >
> > > Actually, the real-time control or dynamic tasks the reviewer mentioned seem more like works from a different area. They are not the core issue of our research so our method cannot handle these tasks.
> >
> > I understand that this might simply be a limitation of the approach, but I stand by my words that clearly communicating such assumptions and limitations to readers is important. I appreciate the effort that the authors have made to clarify these in their rebuttal and revised manuscript. I believe that my main concerns have been addressed. Given that I lean more positively than my fellow reviewers, I will wait for them to engage before making my final judgement.

---

> > > ### Author Response · Authors · 2024-11-22
> > >
> > > We greatly appreciate your response. We agree that these assumptions and limitations are important to readers so we added them to **Limitation and Future Work**. Besides, we are glad to hear that your current concerns have been resolved. If you have further concerns that need to be discussed with us, we are willing to discuss and provide as many details as possible.

---

> > > > ### Comment · Reviewer_UhwZ · 2024-11-26
> > > >
> > > > After further deliberation and in response to the authors' discussion with other reviewers, I have increased my score from 6 to 8 and vote for acceptance.

---

> > > > > ### Author Response · Authors · 2024-11-27
> > > > >
> > > > > Thank you very much for your thoughtful review. We truly appreciate your valuable feedback, which has helped improve our paper. Your support means a lot to us.

---

### Official Review · Reviewer_E4Wn · 2024-11-04

**Soundness:** 3
**Presentation:** 2
**Contribution:** 3
**Rating:** 6
**Confidence:** 4

**Summary:**

The paper proposed an offline-online safe RL framework based on safedreamer, integrating the concepts from RecoverRL. When feasible, the method optimizes rewards, and when infeasible, it focuses on cost optimization. Additionally, the reachability estimation function from RESPO is utilized to introduce cost into the optimization process. To mitigate the issue of inaccurate critic estimates during offline training, in-sample actions from Implicit Q-learning are employed in the Q function learning.

**Strengths:**

1. The proposed safe model-based RL framework effectively addresses offline-online generalization tasks.
2. It demonstrates the capability to safely fine-tune in previously unseen safety-constrained scenarios during real-world deployment.

**Weaknesses:**

1. In Figure 4, the results for DreamerV3 could be omitted, as they overshadow the cost performance of all baseline methods.
2. There are too few obstacles in the real-world environment, making it difficult to assess the agent's obstacle avoidance behavior. And the website does not provide any differences between the proposed algorithm and the baseline in the video demos.
3. In Figure 4, why do the rewards for FOSP in PointGoal2 and PointGoal1 not continue to rise? Additionally, SafeDreamer does not show an increase in reward for PointButton1. Does this indicate that the fine-tuning phase was ineffective? According to the paper's description, the offline data comprises a mixture of unsafe, safe, and random policies; thus, the policy trained on offline data should not be optimal, and performance should improve during the online fine-tuning phase.

**Questions:**

1. I'm curious about the video demonstration of the performances of SafeDreamer and FOSP in the SafetyFadingEasy and Hard environments. Based on the experimental results you presented, FOSP seems to perform better than SafeDreamer. This environment tests the agent's memory, yet it appears that FOSP does not specifically address this aspect. What accounts for the performance improvement?
2. Why do you need to learn Q(s, a) when I recall that DreamerV3 only uses V(s)?
3. How did you adapt Recovery RL to the image-based setting, and how did you modify SafeDreamer to the offline setting?

---

> ### Author Response · Authors · 2024-11-20
>
> We are very grateful to the reviewer for pointing out some of the weaknesses it considers. We arewilling to respond to each of the weaknesses and also answer the reviewer's questions one by one.
>
> **Weaknesses:**
>
> **1. In Figure 4, the results for DreamerV3 could be omitted.**
>
> We appreciate the reviewer’s suggestion. We have included the omitted images in Appendix **Figure 14**.
>
> **2. There are too few obstacles in the real-world environment, making it difficult to assess the agent's obstacle avoidance behavior. The website does not provide any differences between the proposed algorithm and the baseline in the video demos.**
>
> We admit that real-world experiments are relatively simple. However, it is hard for us to add obstacles to the experiment because the robotic workspace is limited. Since its end effector is big enough, we need to leave enough space in the workspace for it to perform tasks. Besides, because of the limited height of the workspace, the robotic arm needs to plan a complex trajectory to avoid collision obstacles. Therefore, to some extent, it reflects the method’s ability to avoid obstacles. The website has showcased videos comparing the proposed algorithm with the baseline. The differences are the same as Figure 12 in the paper. The baseline encounters issues with constraint violations and planning difficulties, while our approach effectively completes the tasks.
>
> **3. Some questions in Figure 4.**
>
> Figure 4 (Figure 3 in the new version) has been smoothed so it seems like FOSP does not show improvement in PointGoal2 and PointGoal1.
>
> Here are some evaluation results about FOSP fine-tuning on PointGoal2 and PointGoal1:
>
> |Environments|Reward||||||Cost||||||
> |----|----|----|----|----|----|----|----|----|----|----|----|----|
> |**Fine-tuning Steps**|0|20k|50k|100k|300k|500k|0|20k|50k|100k|300k|500k|
> |PointGoal1|18.723|18.692|19.581|20.063|19.649|21.517|1.451|3.4|2.719|0.328|0.268|0.2|
> |PointGoal2|8.1|10.31|10.157|12.446|13.034|13.556|7.556|1.486|0.427|0.18|0.214|0.234|
>
> The results indicate that in these two environments, there is indeed a slight increase, though not substantial. The cost also has been decreased. So, it still highlights the importance of fine-tuning in FOSP. As for the reason why the improvement isn't very significant, it is true that the policy trained on offline data is not optimal. But PointGoal1 and PointGoal2 are relatively simple compared to other tasks. Thus, it is possible that the policy can achieve near-optimal performance in relatively simple environments by relying solely on offline data.
>
> As for the experiments for SafeDreamer in PointButton1, although there was no significant improvement in performance, the fine-tuning phase still helped reduce the cost:
>
> |Fine-tuning Steps|0|50k|100k|300k|500k|
> |----|----|----|----|----|----|
> |Reward|8.963|10.021|11.249|8.465|10.754|
> |Cost|10.234|6.319|7.596|3.032|4.549|
>
> **Questions:**
>
> **1. About the videos of the SafetyFadingEasy1 and Hard1**
>
> The videos are presented at https://anonymous.4open.science/r/PointFading_video-F34F
> In a way, these two environments indeed tested the agent's memory capabilities. The model (Dreamerv3, which our method is based on) does possess some memory capability. The deterministic part of the dynamic model allows it to remember information over many time steps [1]. However, relying solely on the model's memory capacity to generalize across these two environments is challenging (SafeDreamer, which includes a similar design, does not perform as well as our method, as shown in Figure 17). We should focus on the increasing of task difficulty. During the fine-tuning process, with the disappearance of the goals, the agent needs to quickly find a path to reach the goal. And our method can quickly capture this ability. We have added it to **Appendix H.3**.
>
> **2. Why do we use $Q(s,a)$ instead of $V(s)$?**
>
> It is common in reinforcement learning that use state value function $V(s)$ and action-state value function $Q(s,a)$ to optimize the problem. The differences between them are $V(s)$ can evaluate which states are more desirable but not directly decide the action and $Q(s,a)$ can directly choose which action to take in any state by selecting the action with the highest value. These two approaches are equivalent since $V^*(s) = \max_{a} Q^*(s,a)$ (* represents optimal). While Dreamerv3 uses $V(s)$, it only considers state value to adjust its policy. In contrast, we leverage $Q(s,a)$ as the critic to focus on the state-action pairs’ value and directly optimize the policy.
>
> **3. About the image-based setting adaptation for Recovery RL**
>
> Please refer to the part on Recovery RL in **Appendix F**. We use the same method as [3] by using a VAE-based model to capture the high-dimensional information.

---

> ### Author Response · Authors · 2024-11-20
>
> **4. About the modification of SafeDreamer for offline setting**
>
> We do not modify it to the offline setting because it will become a new algorithm if we design it as an offline algorithm. Our experiments provide a straightforward validation that our method outperforms SafeDreamer. Additionally, the reviewer may ask how we can be certain whether the improvement in FOSP's performance is solely due to the offline design (since no offline modifications were made to SafeDreamer), and how the other design components contribute. We have answered this question and conducted ablation studies in Section 5.3 to demonstrate that each module contributes to FOSP's performance. Given that FOSP is built upon SafeDreamer, we can confidently assert that each module in our algorithm is necessary for its effectiveness.
>
> References:
>
> [1] Hafner et al., "Learning Latent Dynamics for Planning from Pixels", 2018.
>
> [2] Kostrikov et al. "Offline reinforcement learning with implicit q-learning." arXiv preprint arXiv:2110.06169 (2021).
>
> [3] Brijen et al. “Recovery rl: Safe reinforcement learning with learned recovery zones.” IEEE Robotics and Automation Letters, 6 (3):4915–4922, 2021.

---

> > ### Comment · Reviewer_E4Wn · 2024-11-21
> >
> > If SafeDreamer was not modified for the offline setting, how was the training curve in Figure 3 obtained? You mention using SafeDreamer to collect data and pretrain FOSP, yet the SafeDreamer curve is trained from scratch. This suggests that the performance improvement may be attributed to the use of additional data, rather than the proposed method itself. To ensure a valid evaluation, SafeDreamer should also be pretrained using the same dataset.

---

> > > ### Author Response · Authors · 2024-11-21
> > >
> > > We appreciate the review’s comments. We may not have fully understood the reviewer's point earlier. When we say “SafeDreamer was not modified for the offline setting”, we mean that the SafeDreamer algorithm itself was not adapted or changed specifically for the offline setting. However, it did undergo offline pretraining. In the offline setting, the offline dataset is initialized as the replay buffer of SafeDreamer so it only needs to sample the trajectories from the dataset without environmental interaction. During the online fine-tuning, SafeDreamer first initializes the dataset by the offline dataset, it then interacts with the environment to collect data into the dataset. (Please refer to Appendix A.1 in [1])
> > >
> > > Specifically, in [1] Appendix A.1 Algorithm 2, we modify line 2 to "Initialize offline dataset" and remove line 14 to enable offline training (line 13 is unnecessary in the SafeDreamer BSRP-Lag version).
> > >
> > > In Figure 3, FOSP, SafeDreamer, DreamerV3, and Recovery RL all undergo offline pretraining and online fine-tuning. Figure 3 presents the online fine-tuning part. Other dashed lines represent the online-only results.
> > >
> > > We have updated a new version to clearly show the cost performance of the main baselines in Figure 14. Meanwhile, we add the discussion of SafeDreamer to Appendix F.
> > >
> > > References:
> > >
> > > [1] Huang, Weidong, et al. "Safe dreamerv3: Safe reinforcement learning with world models." arXiv preprint arXiv:2307.07176 (2023).

---

> > > > ### Comment · Reviewer_E4Wn · 2024-11-26
> > > >
> > > > Thanks to the author for addressing most of my concerns. I've improved my score.

---

> > > > > ### Author Response · Authors · 2024-11-27
> > > > >
> > > > > Thank you for taking the time to reassess our work. We greatly appreciate your acknowledgment of our efforts to address your concerns and we are glad the improvements align with your expectations.

---

> ### Comment · Reviewer_E4Wn · 2024-11-21
>
> The Figure 14 still overshadow the cost performance of the main baselines(eg. safedreamer, recovery RL).

---

### Official Review · Reviewer_gtcR · 2024-11-04

**Soundness:** 2
**Presentation:** 2
**Contribution:** 3
**Rating:** 6
**Confidence:** 4

**Summary:**

This paper proposed a model-based offline-to-online safe RL algorithm, FOSP. The proposed framework first uses an offline dataset to learn a world model and a pre-trained policy. The pre-trained policy is then fine-tuned through online safe RL fine-tuning. The effectiveness of FOSP is demonstrated through experiments in the Safety gym and a real-world robot experiment.

**Strengths:**

Extensive experiments and ablation studies were conducted, including a real-world robot experiment with high-dimensional visual observations. The proposed FOSP algorithm achieved good performance, especially in the real-world robot experiments, and outperformed baseline approaches.

**Weaknesses:**

While the proposed algorithm performed well in extensive experiments, I found it hard to appreciate its technical contributions due to its complex algorithm design (e.g., many moving parts, iterations between offline and online learning) and confusing presentation. It is unclear to me if all the design choices are necessary and which components are novel or come from the literature. It would be good if the authors could clearly state their technical contributions and novelty (e.g., does the novelty mainly lie in a new combination of existing components in the literature?). It would also help if the authors could provide more insights on why FOSP would outperform baseline approaches, especially SafeDreamer.

**Questions:**

1. Where does the behavior policy $\pi_b$ in Sec. 4.2 come from? Does it refer to the offline RL policy from Sec. 4.1?
2. Does the training step in Sec. 4.2 also occur offline? Why is it necessary to divide the offline learning stage into two steps?
3. The derivation in Sec. 4.2 is very hard to follow. What is the advantage of using the reachability estimation function from RESPO? It would be helpful to introduce RESPO in the Preliminaries Section. Is $1(s\in S_f)$ equivalent to $u^\pi(s)$? How is Eqn (14) derived from Eqn (13)? Why would introducing advantage functions simplify the constraints?
4. The limitation and future work section should be moved to the main text.

---

> ### Author Response · Authors · 2024-11-20
>
> We are very grateful to the reviewer for pointing out some of the weaknesses it considers. We are willing to respond to each of the weaknesses and also answer the reviewer's questions one by one.
>
> **Weaknesses**:
>
> **1. Confusing presentation**
>
> We are sorry about the confusion. This paper intends to train a vision-based safe policy that can leverage offline data and quickly adapt to novel environments while maintaining safety properties. It’s a non-trivial problem because real-world scenarios often differ significantly from the exact conditions represented in the dataset. Prior works either struggle with limitations in offline learning or face challenges in safe policy fine-tuning. Our main contribution is proposing a new framework to solve this problem. The most promising previous work is SafeDreamer, which is a strong vision-only model-based safe RL algorithm. However, SafeDreamer has limitations in offline learning. First, it suffers from value overestimation that will result in an inaccurate value estimation. Second, its policy learning approach struggles to handle offline constraints (i.e., the policy needs to stay close to the behavior policy) and safety constraints simultaneously, leading to suboptimal task performance and safety trade-offs. Third, SafeDreamer is unable to safely generalize its offline trained policy during online fine-tuning. To address these problems, we propose several solutions building upon SafeDreamer. We first leverage IQL [1] to address the value overestimation and make sure it can learn an accurate Q-value. To ensure safety while not degrading performance, we introduce the reachability estimation function from RESPO [2] and derive a new safe reinforcement learning algorithm that can be adapted to our framework. Inspired by PEX [3], we propose a safe adaptation of its methodology, enabling robust fine-tuning. These components are seamlessly integrated into a unified framework, ensuring compatibility and synergy among the methods. The experiments demonstrate that the proposed method can solve safe offline-to-online and safe generalization tasks. Our framework is novel and we made some technical contributions to the algorithm (e.g., deriving safe RL based on REF, a safe version of PEX) to ensure they can solve our problem.
>
> **2. Why would FOSP outperform SafeDreamer**
>
> SafeDreamer is an online safe model-based RL algorithm. It is used to improve vision-based tasks in safe RL. Our method is better than SafeDreamer according to the analysis in Weakness 1. Furthermore, we can analyze the outperforming through experiments from two perspectives:
>
> 1) The offline-to-online tasks: we use the same method to train FOSP and SafeDreamer, i.e., offline to online (Table 1). SafeDreamer suffers from the distribution shift in the offline setting and it cannot well handle the constraints during offline training. So, it cannot perform well at the initial stage of fine-tuning, which would hinder its subsequent fine-tuning. Meanwhile, SafeDreamer lacks an offline-to-online design, making it difficult to fine-tune the RL policy.
>
> 2) The online tasks: we conduct online training only for SafeDreamer, while FOSP undergoes offline-to-online training (Table 2). For a fair comparison, FOSP's online training steps are the same as SafeDreamer's online training steps (just like in the real world, offline training does not require any cost). When SafeDreamer trains only online, it lacks the safety guidance from the offline dataset. It also struggles with step limits as it always requires a large number of training steps to achieve optimal performance [4]. In contrast, FOSP can take advantage of both the offline dataset and the fine-tuning phase.
>
> It is why FOSP outperforms SafeDreamer.
>
> **3. Are all the design choices necessary?**
>
> Yes, the experimental results have been shown in Section 5.3 and Figure 3 (Figure 4 in the new version).
>
> **Questions**:
>
> **1. About the behavior policy**
>
> The definition of the behavior policy can be found in “Offline Reinforcement Learning: Tutorial, Review, and Perspectives on Open Problems” [5] Figure 1. Precisely, behavior policy is the policy used when collecting datasets. It is not the offline RL policy. In our experiment, we used three kinds of policies (safe, unsafe, random) to collect datasets so the behavior policy could be a mixture of these three policies.
>
> **2. The training step in Section 4.2**
>
> To help the reviewer better understand the method, please see the pseudo-code (Appendix C). Section 4.2 provides a safe RL algorithm, which can be used offline and online to ensure safety. We divided the offline training phase into Sections 4.1 and 4.2, as Section 4.1 primarily addresses value estimation, while Section 4.2 introduces a method for training a safe policy based on the critic training approach outlined in Section 4.1. Dividing into two parts can help readers to understand how we solve different problems in different ways.

---

> ### Author Response · Authors · 2024-11-20
>
> **3. The reachability estimation function in Sec. 4.2**
>
> We appreciate the suggestion to introduce the RESPO in the Preliminaries. The advantage of using the reachability estimation function is to ensure the policy can learn how to reenter the feasible region when the agent is in the infeasible region (shown in Figure 2). It can help the policy learn more effectively and balance the trade-off between its performance and cost (Figure 3 (Figure 4 in the new version) demonstrates the advantage of REF). $u^{\pi}(s)$ do not equal to $1(s \in S_f)$ as it is actually an expectation. We would like to thank the reviewer for pointing out it since this definition could be difficult to understand. So we replace it with $ u^\pi(s) = \mathbb{E}\_{\tau\sim\pi}[\max\_{s\_t\in \tau}(1 - 1((s_t|s_0, \pi)\notin\mathcal{S}_f))]$. The value in the expectation is **equal to 0 if there exists an infeasible state and 1 if all states are in the feasible region in a specific trajectory $\tau$**. When the current policy encounters a local infeasible region, the reachability estimation function can help it reenter the feasible region from the edge of infeasible regions and ensure all the states in this trajectory are safe.
>
> **4. The derivation in Sec. 4.2**
>
> The deviation is obvious and introduced by prior work [2]. The reviewer can find it on page 5. Briefly, it first converts equation 12,13 (13, 14 in the new version) to
> $$\max\_{\pi\_\psi} \mathbb{E}\_{s} [ V\_\varphi^r (s) \cdot 1 (s \in S_f)- V_\varphi^c (s)\cdot 1 (s \notin S_f) ], \text{s.t.}  V\_\varphi^c (s) = 0, D_{\mathrm{KL}}(\pi_\psi||\pi_b) \leq \epsilon.$$
> Then, we can get equation 14 (15 in the new version).
>
> Equation 14 (15 in the new version) shows the function of the reachability estimation function. Equation 15 (16 in the new version) is the optimization problem and the deviation from equations 12,13 (13, 14 in the new version) to equation 15 (16 in the new version) is in Appendix A. The logic could be $Eq.13,14 \rightarrow Eq.15, Eq.13,14 \rightarrow Eq.16$ (numbers follows the new version).
>
> **5. Why would introducing advantage functions simplify the constraints?**
>
> It is ambiguous that introducing advantage functions does not simplify the constraints. Instead, introducing the Augmented Lagrangian can simplify the constraint as it can transform constrained optimization into unconstrained optimization. For the advantage function, we aim to capture the influence of actions within this optimization problem. A larger value function does not guarantee the existence of an action that will lead the agent to the desired state, so it's crucial to account for action effects. Proposition 1 provides a straightforward way to integrate the advantage function into this optimization framework. We have revised it in the paper.
>
> **6. The limitation and future work section should be moved to the main text.**
>
> We thank the reviewer for the suggestion. Due to space limitations, we cannot move them all to the main body. Therefore, we simplified them in the **Conclusion** and guided readers to Appendix I.
>
> References:
>
> [1] Kostrikov, Ilya, Ashvin Nair, and Sergey Levine. "Offline reinforcement learning with implicit q-learning." arXiv preprint arXiv:2110.06169 (2021).
>
> [2] Ganai, Milan, et al. "Iterative reachability estimation for safe reinforcement learning." Advances in Neural Information Processing Systems 36 (2024).
>
> [3] Zhang, Haichao, We Xu, and Haonan Yu. "Policy expansion for bridging offline-to-online reinforcement learning." arXiv preprint arXiv:2302.00935 (2023).
>
> [4] Huang, Weidong, et al. "Safe dreamerv3: Safe reinforcement learning with world models." arXiv preprint arXiv:2307.07176 (2023).
>
> [5] Levine, Sergey, et al. "Offline reinforcement learning: Tutorial, review, and perspectives on open problems." arXiv preprint arXiv:2005.01643 (2020).

---

> ### Comment · Reviewer_gtcR · 2024-11-23
>
> Thank the authors for their detailed response. After reading the response, I still have two questions/comments:
>
> 1. Algorithm overview: ideally, one should be able to easily follow the paper without referring to the appendix (e.g., pseudo-code). To help the readers better understand the connections between Sec. 4.1 and Sec. 4.2, the authors may consider adding a concise version of the algorithm in the main text. Alternatively, the authors may describe the algorithm in the text at the beginning of Sec. 4. In particular, it would be helpful to formally define the training objective of the offline training stage before delving into Sec. 4.1 and 4.2 in detail.
>
> 2. I still have questions regarding the reachability estimation function and related derivations in Sec. 4.2. I have checked the original definitions and derivations in [2]. If I understand correctly, $1(s \in S_f)$ is the notation of the optimal feasible set in RCRL where the environment is deterministic. REF is a probabilistic notation of the optimal feasible set applicable when the environment is stochastic. Therefore, I get confused about how Eqn. (13) and Eqn. (14) can lead to Eqn. (15), given that they adopt different assumptions of the environment. Also, I believe the set $\mathcal{S}_f$ in $u^\pi(s)$ should be $\mathcal{S}_v$ instead, i.e., the constraint violation set. It is different from the optimal feasible set $S_f$.

---

> ### Author Response · Authors · 2024-11-23
>
> We greatly appreciate the reviewer’s suggestion. We are willing to revise the paper and make some explanations to solve his concerns.
>
> **1. About the algorithm overview**
>
> We agree that explaining the algorithm in Section 4 would help readers to easily follow our work. We have updated the paper and revised the beginning of Section 4. In the new version, we mention the function of Section 4.1 and Section 4.2 to clarify their connection. Meanwhile, we emphasize the training objectives, with Section 4.1 detailing the Critic training and Section 4.2 focusing on the safe Actor training. We also provide guidance for readers in finding pseudo-code.
>
> **2. About the reachability estimation function**
>
> It is true that the RESPO can be used when the environment is stochastic. However, it also can be used when the environment is deterministic. The algorithm was defined in the deterministic case in [1] Section 5.1, and the stochastic case in Section 5.2 so the REF can be used in both cases. Thus, Eqn. (13) and Eqn. (14) can lead to Eqn. (15). In addition, it has two advantages when compared with RCRL (Section 4.2 in [1]):
> 1) By optimizing $V_h$ and $V$ simultaneously using probabilities rather than separately with indicator functions, it ensures the policy can (re)enter the feasible set, making it easier to achieve the optimal;
> 2) It is not limited to deterministic MDPs and can be applied to stochastic environments.
>
> We want to leverage the first advantage to solve our problem. Plus, it is also good at handling mixed constraints according to Section 6.2 which we have mentioned in our paper.
>
> For the definition of $u^{\pi}(s)$, we are so sorry that we found that our previous response was mistaken and we have corrected it. Our definition of $u^{\pi}(s)$ should have this form: $u^\pi(s) = \mathbb{E}\_{\tau\sim\pi}[1 - \max_{s_t\in \tau}(1\{(s_t|s_0, \pi)\in S_v\})]$. The value in the expectation is **equal to 0 if there exists an infeasible state and 1 if all states are in the feasible region** in a specific trajectory $\tau$. And our optimization can be written as equation 15:
>
> $$\max_{\pi} \mathbb{E}[V^r(s)\cdot u^{\pi}(s) – V^c(s)\cdot (1-u^{\pi}(s))], s.t. V^c(s) = 0$$
>
> In contrast, the RESPO uses the definition of $ u^{\pi}(s) := \mathbb{E}\_{\tau\sim\pi} [\max_{s_t\in \tau}1((s_t|s_0, \pi)\in S_v)]$ since its optimization problem is
> $$\max_{\pi} \mathbb{E}[V^r(s)\cdot (1-u^{\pi}(s)) – V^c(s)\cdot u^{\pi}(s)], s.t. V^c(s) = 0$$
> (refer to [1] equation 3).  The value in its expectation is **equal to 1 if there exists an infeasible state and 0 if all states are in the feasible region**. In summary, our definition is just an “opposite version” of RESPO’s definition so they are the same.
>
> References:
>
> [1] Ganai, Milan, et al. "Iterative reachability estimation for safe reinforcement learning." Advances in Neural Information Processing Systems 36 (2024).

---

> > ### Comment · Reviewer_gtcR · 2024-11-28
> >
> > Thank the authors for addressing my comments. I have raised my score.

---

> > > ### Author Response · Authors · 2024-11-28
> > >
> > > Thank you very much for your valuable suggestions regarding our work. We are glad that our work can be further improved with your input. We truly appreciate your active responses during the discussion process!

---

### Official Review · Reviewer_2vUt · 2024-11-04

**Soundness:** 3
**Presentation:** 3
**Contribution:** 2
**Rating:** 8
**Confidence:** 4

**Summary:**

This paper presents FOSP, a novel safe reinforcement learning method. FOSP primarily addresses the issue of enhancing safety through offline training and online fine-tuning. Its main contributions include:

1. The introduction of an offline-to-online reinforcement learning framework.
2. Enhanced safety and performance balance during visual tasks through the integration of offline and online phases.
3. In real-world deployments, FOSP allows safe fine-tuning in unseen safety constraint scenarios.
4. The experiments validate the method's effectiveness in both simulated and real robotic tasks and demonstrate its safe generalization ability in new scenarios.

**Strengths:**

1. The FOSP method introduces the concept of safe generalization within the offline-to-online reinforcement learning framework, combining world models with offline training and online fine-tuning to enhance safety and performance. This novel strategy for applying reinforcement learning in safety-critical scenarios is insightful.
2. The design of experiments is comprehensive while covering tasks in both simulated environments and validations in real robotic settings.
3. The experiments of FOSP in real robotic tasks demonstrate real-world value and safe generalization in unseen scenarios.
4. The paper clearly articulates the optimization objectives at each stage, aiding in understanding.

**Weaknesses:**

1. While the experiments were validated in Safety-Gymnasium and real robotic environments, the number and complexity of tasks remain relatively limited.
2. The motivation for introducing the new offline to online setting in the paper has not been well communicated.

**Questions:**

1. Although this method performs better than others, it also introduces a new design for online to offline transitions. What specific problem is this design intended to address? The reviewer hopes the authors can clarify the benefits of this setting. For instance, the reviewer notes that in Table 1, the performance of SafeDreamer (offline) is actually comparable to that of FOSP.
2. How is safety ensured in this method, particularly when dealing with unseen data and new environments? Specifically, how does the online fine-tuning process avoid safety hazards? For instance, how does the constraint violation during online fine-tuning compare to the converged offline policy?
3. The reachability estimation function is a key component of this paper. How is the feasible part determined through $ S_f(\boldsymbol{s}):=\{\boldsymbol{s}|V_\varphi^c(\boldsymbol{s})=0\} $? In particular, how does this hold when $ V_{\varphi}^c(\boldsymbol{s})=0 $ may not strictly hold in the fitted value function?
4. How do the authors view the potential of this method for safe sim2real applications? That is, training under simulated environment data while ensuring safety and improving performance during real-world deployment.
5. The safety policy expansion mechanism stabilizes performance during the initial fine-tuning phase, but does it impose limitations on long-term fine-tuning? For example, what would the comparison of results be between the FOSP(online) version and FOSP in short-term versus long-term training?

**Details Of Ethics Concerns:**

No ethics concerns.

---

> ### Author Response · Authors · 2024-11-20
>
> We are very grateful to the reviewer for pointing out some of the weaknesses it considers. We are very willing to respond to each of the weaknesses and also answer the reviewer's questions one by one.
>
> **Weaknesses:**
> **1. While the experiments were validated in Safety-Gymnasium and real robotic environments, the number and complexity of tasks remain relatively limited.**
>
> The task complexity is indeed a limitation in our paper, especially for real-world experiments. However, few works have applied safe reinforcement learning to image-based robotic arm planning tasks. Therefore, migrating it to the real world is a non-trivial step, although the task is relatively simple. We have already written this in our future work in **Appendix I**. For the simulation environment, we selected a more complex environment to validate our method: Race2. We illustrate our experimental results as follows:
>
> ||Reward|Cost|
> |----|----|----|
> |Safedreamer|2.1 $\rightarrow$ 4.3|20.6 $\rightarrow$ 2.5|
> |FOSP|2.5 $\rightarrow$ 4.4|8.8 $\rightarrow$ 0.6|
>
> The results indicate that our method is still better than baselines in more complex environments. We have updated them in **Appendix H.6**.
>
> **2. The motivation for introducing the new offline to online setting in the paper has not been well communicated.**
>
> We appreciate the reviewer for mentioning it. This new setting involves safe offline and online tasks. Here is our motivation:
>
> We want to train a safe policy on an offline dataset and enable it to generalize to similar unseen tasks. The most straightforward idea is to use offline safe RL to train the policy. However, offline RL suffers from distribution shift so the offline trained safe policy struggles to generalize to similar safety-critical tasks with nearly-zero constraint violations. The distribution shift will make it harder to deal with unseen constraints in the new environment. Directly applying it to real-world experiments can lead to severe constraint violations. To address this, we propose a new framework FOSP to leverage the knowledge from offline datasets and enhance performance through fine-tuning. It shows that our method can perform well during offline pretraining and improve tasks’ performance while ensuring constraints during online fine-tuning. Furthermore, it can solve some unseen tasks with near-zero violations.
>
> This part has been added to the **Introduction**.
>
> **Questions:**
>
> **1. What specific problem is this method intended to address? The reviewer hopes the authors can clarify the benefits of this setting. For instance, the reviewer notes that in Table 1, the performance of SafeDreamer (offline) is actually comparable to that of FOSP.**
>
> We aim to solve safe generalization tasks. The offline-to-online framework is a common structure for solving real-world tasks. In our paper, we consider safe offline-to-online settings. We need to learn as much knowledge as possible from offline datasets and ensure relative safety of interaction with the environment during online fine-tuning. Safe online fine-tuning can further teach the model how to safely complete unseen but similar tasks. Therefore, its generalization can be improved on some safety-critical tasks that have not been seen before.
>
> The benefits of our method are shown as follows:
>
> 1) It can learn better performance with lower cost from the dataset during the offline phase (shown in Figure 13).
> 2) It can improve performance by online fine-tuning while ensuring near-zero constraint violations (shown in Figure 4 (Figure 3 in the new version)).
> 3) It can generalize to unseen tasks while maintaining the cost as low as possible (shown in Figure 6, Figure 17, Figure 12).
>
> In Table 1, the cost of SafeDreamer (offline) is comparable to that of FOSP (offline). However, with the same cost, FOSP (offline) can achieve better performance, which is its benefit.
>
> **2. How is safety ensured in this method, particularly when dealing with unseen data and new environments? Specifically, how does the online fine-tuning process avoid safety hazards?**
>
> The safety of fine-tuning can be guaranteed by Section 4.3. Specifically, we train a new policy online and let it follow the guidance of the offline trained policy. During the exploration, we choose the action from these two policies according to their safety. So, at the beginning of the fine-tuning, the offline-trained policy can recognize some hazards similar to those in the original environment and avoid them. But it could be challenging for it to handle some novel hazards. As fine-tuning progresses, the online policy gradually learns the new constraints and further improves its performance, enabling it to handle novel hazards. We compare the converged offline FOSP and FOSP during online training. The results are shown as follows:

---

> ### Author Response · Authors · 2024-11-20
>
> |Metrics|Fine-tuning Steps|PointButton1|PointGoal1|PointGoal2|PointPush1|CarGoal2|Transfer(Goal1$\rightarrow$Goal2)|
> |----|----|----|----|----|----|----|----|
> |**Reward**|offline converged|14.65|18.723|8.1|4.092|10.146|18.723(Original environment), 3.74(New environment)|
> ||20k|13.415|18.692|10.31|7.253|9.063|7.082|
> ||50k|13.569|19.581|10.157|7.916|9.816|9.141|
> ||100k|15.306|20.063|12.446|9.148|10.275|10.838|
> ||300k|17.861|19.649|13.034|10.373|12.378|11.503|
> ||500k|18.102|21.517|13.566|13.281|14.512|12.116|
> |**Cost**| offline converged |9.622|1.451|7.556|18.117|1.618|1.451(Original environment), 4.916(New environment)|
> ||20k|8.639|3.4|1.486|1.232|1.063|1.399|
> ||50k|6.664|2.719|0.427|1.497|0.385|0.561|
> ||100k|4.866|0.328|0.18|1.089|0.394|0.045|
> ||300k|4.256|0.268|0.214|0.507|0.18|0.272|
> ||500k|2.188|0.2|0.234|0.157|0.071|0.028|
>
> We found that the cost may slightly increase at the beginning of the fine-tuning in the transfer task. However, it will decrease fast during fine-tuning. So online fine-tuning could be effective to avoid hazards. Furthermore, we found that fine-tuning helps reduce costs even in the same environment. This is because, due to dataset limits, the policies trained offline are often suboptimal, and online fine-tuning can further enhance its safety. We have added the results to **Appendix H.3**.
>
> **3. The reachability estimation function questions:**
>
> Explanation: Let’s back to the preliminaries (Safe Model-based RL). The cost functions $\mathcal{c}: \mathcal{S} \times \mathcal{A}\rightarrow [0,C_{max}]$. It means the cost values are non-negative, which ensures that $V^c(s)$ := $\mathbb{E}\_{\tau \sim \pi}$ $[ \sum\_{t=0}^{T} \gamma^t c(s\_t, a\_t) ]$ are non-negative. Therefore, we have the following inference [1]:
> - $V^c(s) = 0 \Rightarrow \forall s_t, c(s_t) = 0$, so the policy $\pi$ can satisfy the constraints starting from $s$. Moreover, the optimal cost value function $V^c(s) = 0 \Rightarrow \exists \pi, \min_{\pi}V^c = 0,$ meaning that a policy that satisfies the constraint exists.
> So the fitted value function equal to zero can always hold only if all states that the policy passes through are safe. The feasible part can be determined by $S_f(s)$ = {$s|V^c(s)=0$}.
>
> **4. How do the authors view the potential of this method for safe sim2real applications? That is, training under simulated environment data while ensuring safety and improving performance during real-world deployment.**
>
> Of course, our framework has the potential for sim2real applications. Although we show the experiments in the offline-to-online setting in this paper, its potential in safe sim2real is undeniable. In the safe sim2real setting, the model should be first trained in the simulation so it does not need to be trained offline. After training in the simulation, it can achieve relatively good performance and have some safety perceptions. Then, it needs to be fine-tuned in the real world. World models are data-efficiency, so they will be effective in using trajectories collected from the real world. Therefore, it can gradually handle this task through a small number of attempts. Additionally, safe fine-tuning can ensure it will perform as safely as possible during the fine-tuning phase. Finally, the model can transfer from simulation to the real world.
>
> We have added it into **Future Work** to provide some insights for the community.
>
> **5. The safety policy expansion mechanism stabilizes performance during the initial fine-tuning phase, but does it impose limitations on long-term fine-tuning? For example, what would the comparison of results be between the FOSP(online) version and FOSP in short-term versus long-term training?**
>
> It won’t hinder the long-term fine-tuning. The offline trained policy is often suboptimal and it has been frozen during the online fine-tuning. Hence, the new online policy will outperform the offline trained policy as long as the fine-tuning steps are sufficient. Meanwhile, thanks to the data efficiency of the world model, the new online policy can converge quickly. We compare the FOSP(directly online) and FOSP with short-term and long-term fine-tuning (offline pretrained 1M steps). The results are shown as follows:
>
> |Model tpye|Training Steps|Metric|PointButton1|PointGoal1|PointGoal2|PointPush1|CarGoal2|
> |----|----|----|----|----|----|----|----|
> |FOSP(online)|500k|Reward|10.57|14.749|9.471|1.397|8.749|
> |||Cost|6.88|2.66|4.78|2.49|2.12|
> ||1000k|Reward|12.81|18.586|10.717|4.181|10.144|
> |||Cost|2.76|1.08|3.09|2.05|1.21|
> |FOSP(fine-tuning)|50k|Reward|13.569|19.581|10.157|7.916|9.816|
> |||Cost|6.664|2.719|0.427|1.497|0.385|
> ||500k|Reward|18.102|21.517|13.566|13.281|14.512|
> |||Cost|2.188|0.2|0.234|0.157|0.071|

---

> ### Author Response · Authors · 2024-11-20
>
> The results indicate that FOSP(online) does not perform as well as the fine-tuning method, even after training for 1M steps. Direct online training suffers from the number of training steps, usually requiring a large number of steps to converge. The offline pretraining combined with the online fine-tuning framework allows the agent to effectively leverage offline knowledge to improve online performance, with relatively short fine-tuning being sufficient to achieve satisfactory results. This outcome also suggests that long-term fine-tuning is not hindered, and the policy can continue to learn throughout the fine-tuning process. We have added them to **Appendix H.4**.
>
> References:
>
> [1] Yinan Zheng, Jianxiong Li, Dongjie Yu, Yujie Yang, Shengbo Eben Li, Xianyuan Zhan, and Jingjing Liu. Safe offline reinforcement learning with feasibility-guided diffusion model. arXiv preprint arXiv:2401.10700, 2024.

---

> > ### Comment · Reviewer_UhwZ · 2024-11-25
> >
> > Dear reviewer 2vUt,
> >
> > Since the discussion phase is coming to an end, it would be great if you could take a moment to respond to the authors' rebuttal. I'm curious to know whether their response addresses your concerns.

---

> > > ### Comment · Reviewer_2vUt · 2024-11-25
> > >
> > > The reviewer’s concerns have been resolved, and I am pleased to see that the authors have provided thoughtful responses to the insightful questions raised by other reviewers. I am willing to increase my score.

---

> > > > ### Author Response · Authors · 2024-11-25
> > > >
> > > > We sincerely thank you for your kind words. Your constructive feedback has been instrumental in improving the quality of our work and we deeply appreciate your willingness to increase the score.

---

### Author Response · Authors · 2024-11-20
**Paper revise instruction**

We appreciate the suggestions from all the reviewers. The revised version of the paper has been submitted and the revisions are highlighted in red. We would like to write instructions to help reviewers quickly find the corresponding revision.

Main body:
- In Introduction, we add the motivation for introducing safe offline-to-online setting in this paper following advice from Reviewer 2vUt
- In Preliminaries, the introduction of Reachability Estimation Function is added according to Reviewer gtcR’s suggestion.
- In Method, we rewrite the reachability estimation function to make it correspond to the definition. Additionally, we clarify why the method introduces Augmented Lagrangian and advantage function.
- In Conclusion, we state our limitations and guide readers to find the details in Appendix.

Appendix:
- In Appendix H.3, we stress the motivation to transfer our method to Fading environment.
- In Appendix H.3, we add the results of the average reward and cost for the transfer task from SafetyPointGoal1 to SafetyPointGoal2. It indicates that the converged offline policy might violate some unseen constraints in a new environment. And the fine-tuning process can help the agent quickly reduce the violations.
- We illustrate Figure 14, which is a version of Figure 3 with DreamerV3 removed.
- We add the comparison of FOSP (online version) and FOSP to further demonstrate the effectiveness of fine-tuning to Appendix H.4.
- We devise a transfer task with more visual variations in Appendix H.5. We analyze the reason why our method fails to handle the visual change task.
- We add an experiment on a more realistic environment Race in Appendix H.6. We show FOSP has better performance with lower cost even in more complex environments.
- In Limitations, we add the simplicity in the real-world experiment and the failure in novel visual observations. In Future Works, we give some promising solutions to provide the community with more insights into this field. In addition, we give some analysis in safe sim2real following Reviewer 2vUt’s suggestion.

---

### Author Response · Authors · 2024-11-20
**Global response**

We sincerely thank all reviewers for their constructive and thoughtful feedback. The reviewers agreed that the safe offline-to-online method FOSP we proposed is effective in solving the safe generalization problem. We would like to summarize the strengths as follows:

1. The method FOSP uses safe offline-to-online RL to solve safety-critical generalization tasks, which is **novel and insightful**.
2. The experimental design is **comprehensive**, effectively demonstrating that FOSP can safely fine-tune in new environments, showing its strong performance with sufficient baselines.
3. The experiments also include **real-world robotic trials**, making it one of the few works to deploy safe RL into the real world.
4. The paper includes **detailed derivations** of the proposed method, making readers easy to follow.

However, the reviewers had some concerns. We have made every effort to address their concerns by adding detailed explanations and supplementary experiments. The specific points are as follows:

- Reviewer 2vUt, E4Wn and UhwZ are concerned the experiments are **relatively simple**. We have supplemented the Race experiment to show our method can perform well in a more realistic environment. For the real-world experiment, our method has been preliminarily validated for real-world transfer. Some more complex tasks like dynamic obstacles require specific algorithmic designs such as the prediction module, which are not the core of the problem we aim to solve. Thus, we would leave them for future researchers to further explore and we have proposed some promising solutions in the Future Works.
- Reviewer UhwZ wanted to know the impact of **visual changes** on our method. We have conducted an experiment for significant visual changes in the simulation environment. It indicated that our method struggled to handle it. However, it is mainly attributed to the vision-only input of the model rather than our main design. Hence, we propose the addition of semantic information as a possible solution and write it into Future Works.
- Reviewer E4Wn felt confused with some results in **Figure 4** (Figure 3 in the new version). Therefore, we presented the detailed evaluation results in Figure 4 with tables to further demonstrate the effectiveness of the fine-tuning process.
- Reviewer gtcR found it difficult to understand the **technical contributions**. We first clearly clarified our thought process. Then, we used both theory and experiments to explain the importance of the components and why our method outperforms the baselines. Meanwhile, we add more detailed explanations to the paper to help readers better understand our method.
- Reviewer 2vUt thought the **motivation for the new setting** was not well communicated. We have explained the motivations and supplemented them in the paper.

Again, we greatly appreciate all the efforts the reviewers and AC made in providing their suggestions. The reviewers' comments are invaluable for us since they will help us improve our paper.

---

### Meta-Review · Area_Chair_qQRy · 2024-12-21

**Metareview:**

The paper proposes FOSP, an offline-to-online safe RL framework that uses a model-based approach to fine-tune an offline-trained policy. By incorporating in-sample optimization and reachability guidance, FOSP aims to ensure safe performance even in unseen scenarios. Experiments in simulation (Safety-Gymnasium) and real-world tasks demonstrate that FOSP generalizes better and reduces constraint violations compared to SafeDreamer and other baselines.

Strengths:

-- Novel Framework: Presents a new offline-to-online approach to safe RL, integrating model-based updates and safety mechanisms for vision-based tasks.

-- Extensive Experiments: Validates performance in both simulated and real-world settings, underscoring the method’s practical relevance.

-- Real-World Deployment: Demonstrates successful application on real robots, showing promise for real-world safety-critical tasks.

Weaknesses:

-- Limited Task Complexity: Experiments focus on relatively simple tasks; the method’s performance on more complex or dynamic scenarios remains unclear.

-- Algorithm Clarity: The design involves multiple components and iteration steps, making it hard to separate novel elements from existing approaches.

-- Safety Guarantees: More discussion is needed on the method’s robustness to out-of-distribution data or dramatic environment changes.

After carefully reading the paper, the reviews and rebuttal discussions, the AC agrees with the reviewers on recommending to accept the paper.

**Additional Comments On Reviewer Discussion:**

The weaknesses are described above. The authors have addressed most comments in rebuttal and the reviewers generally agree to accept the paper.

---

### Decision · Program_Chairs · 2025-01-22

Accept (Poster)